# Towards Generalizable Reinforcement Learning via Causality-Guided Self-Adaptive Representations

**Yupei Yang[1], Biwei Huang[2]\*, Fan Feng[2,3], Xinyue Wang[2], Shikui Tu[1]\*, Lei Xu[1]**
[1]Shanghai Jiao Tong University, [2]University of California San Diego,
[3]Mohamed bin Zayed University of Artificial Intelligence
{yupei_yang, tushikui, leixu}@sjtu.edu.cn,
{bih007, xiw159}@ucsd.edu, ffeng1017@gmail.com

## Abstract

General intelligence requires quick adaptation across tasks. While existing reinforcement learning (RL) methods have made progress in generalization, they typically assume only distribution changes between source and target domains. In this paper, we explore a wider range of scenarios where not only the distribution but also the environment spaces may change. For example, in the CoinRun environment, we train agents from easy levels and generalize them to difficulty levels where there could be new enemies that have never occurred before. To address this challenging setting, we introduce a *causality-guided self-adaptive representation*-based approach, called CSR, that equips the agent to generalize effectively across tasks with evolving dynamics. Specifically, we employ causal representation learning to characterize the latent causal variables within the RL system. Such compact causal representations uncover the structural relationships among variables, enabling the agent to autonomously determine whether changes in the environment stem from distribution shifts or variations in space, and to precisely locate these changes. We then devise a three-step strategy to fine-tune the causal model under different scenarios accordingly. Empirical experiments show that CSR efficiently adapts to the target domains with only a few samples and outperforms state-of-the-art baselines on a wide range of scenarios, including our simulated environments, CartPole, CoinRun and Atari games.

## 1 Introduction

In recent years, deep reinforcement learning (DRL, (Arulkumaran et al., 2017)) has made incredible progress in various domains (Silver et al., 2016; Mirowski et al., 2016). Most of these works involve learning policies separately for fixed tasks. However, many practical scenarios often have a sequence of tasks with evolving dynamics. Instead of learning each task from scratch, humans possess the ability to discover the similarity between tasks and quickly generalize learned skills to new environments (Pearl & Mackenzie, 2018; Legg & Hutter, 2007). Therefore, it is essential to build a system where agents can also perform reliable and interpretable generalizations to advance toward general artificial intelligence (Kirk et al., 2023).

A straightforward solution is policy adaptation, i.e., leveraging the strategies developed in source tasks and adapting them to the target task as effective as possible (Zhu et al., 2023). Approaches along this line include, but are not limited to, fine-tuning (Mesnil et al., 2012), reward shaping (Harutyunyan et al., 2015), importance reweighting (Tirinzoni et al., 2019), learning robust policies (Taylor et al., 2007; Zhang et al., 2020), sim2real (Peng et al., 2020), adaptive RL (Huang et al., 2021), and subspace building (Gaya et al., 2022). However, these algorithms often rely on an assumption that all the source and target domains have the same state and action space while ignoring the out-of-distribution scenarios which are more common in practice (Taylor & Stone, 2009; Zhou et al., 2023).

---

\*corresponding author

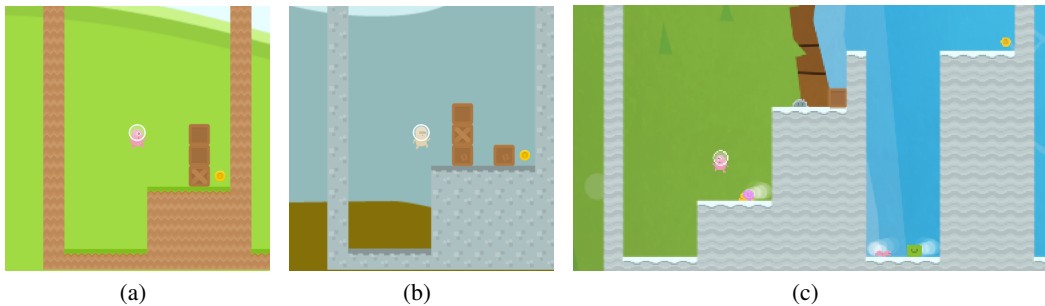

(a)           (b)           (c)

Figure 1: **Environmental changes may or may not necessitate retraining RL agents,** as illustrated on different variations of CoinRun. Changes in the amount and shape of obstacles from (a) to (b) do not prevent the agent from completing the task, while deadly holes and enemies introduced in (c) necessitate retraining.

In this paper, we expand the application of RL beyond its traditional confines by exploring its adaptability in broader contexts. Specifically, our investigation focuses on policy adaptation in two distinct scenarios:

1. *Distribution shifts:* the source and target data originate from the same environment space but exhibit differences in their distributions, e.g. changes in transition, observation or reward functions;

2. *State/Action space expansions:* the source and target data are collected from different environment spaces, e.g. they differ in the latent state or action spaces.

These scenarios frequently occur in practical settings. To illustrate, we reference the popular CoinRun environment (Cobbe et al., 2019). As shown in Fig. 1, the goal of CoinRun is to overcome various obstacles and collect the coin located at the end of the level. The game environment is highly dynamic, with variations in elements such as background colors and the number and shape of obstacles (see Fig. 1(a) and Fig. 1(b)) — this exemplifies distribution shifts. Additionally, CoinRun offers multiple difficulty levels. In lower difficulty settings, only a few stationary obstacles are present, while at higher levels, a variety of enemies emerge and attack the agents (see Fig. 1(c)). To prevail, agents must learn to adapt to these new enemies — this scenario illustrates state/action space expansions.

We propose a **C**ausality-guided **S**elf-adaptive **R**epresentation-based approach, termed CSR, to address this problem for partially observable Markov decision processes (POMDPs). Considering that the raw observations are often reflections of the underlying state variables, we employ causal representation learning (Schölkopf et al., 2021; Huang et al., 2022; Wang et al., 2022) to identify the latent causal variables in the RL system, as well as the structural relationships among them. By leveraging such representations, we can automatically determine what and where the changes are. To be specific, we first augment the world models (Ha & Schmidhuber, 2018; Hafner et al., 2020) by including a task-specific change factor $\theta$ to capture distribution shifts, e.g., $\theta$ can characterize the changes in observations due to varying background colors in CoinRun. If the introduction of $\theta$ can well explain the current observation, it is enough to keep previously learned causal variables and merely update a few parameters in the causal model. Otherwise, it implies that the current task differs from previously seen ones in the environment spaces, we then expand the causal graph by adding new causal variables and re-estimate the causal model. Finally, we remove some irrelevant causal variables that are redundant for policy learning according to the identified causal structures. This three-step strategy enables us to capture the changes in the environments for both scenarios in a self-adaptive manner and make the most of learned causal knowledge for low-cost policy transfer. Our key contributions are summarized below:

- We investigate a broader scenario towards generalizable reinforcement learning, where changes occur not only in the distributions but also in the environment spaces of latent variables, and propose a causality-guided self-adaptive representation-based approach to tackle this challenge.

- To characterize both the causal representations and environmental changes, we construct a world model that explicitly uncovers the structural relationships among latent variables in the RL system.

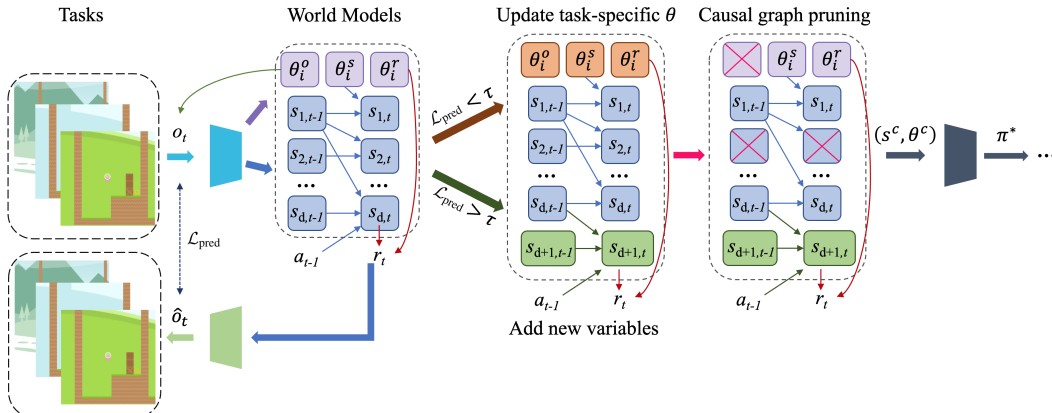

Figure 2: **Efficient policy adaptation through the CSR framework.** For each target task, we first use the prediction error, $\mathcal{L}_{\text{pred}}$, to determine whether it involves distribution shifts or space shifts. We then adjust the model accordingly by updating the task-specific change factor $\boldsymbol{\theta}_i$, or by adding new variables. Finally, we conduct causal graph pruning that removes variables unnecessary for the current task. Based on such compact causal representations, we can efficiently implement policy adaptation in a self-adaptive manner.

- By leveraging the compact causal representations, we devise a three-step strategy that can identify where the changes of the environment take place and add new causal variables autonomously if necessary. With this self-adaptive strategy, we achieve low-cost policy transfer by updating only a few parameters in the causal model.

## 2 WORLD MODEL WITH CAUSALITY-GUIDED SELF-ADAPTIVE REPRESENTATIONS

We consider generalizable RL that aims to effectively transfer knowledge across tasks, allowing the model to leverage patterns learned from a set of source tasks while adapting to the dynamics of a target task. Each task $\mathcal{M}_i$ is characterized by $\langle \mathcal{S}_i, \mathcal{A}_i, \mathcal{O}_i, R_i, T_i, \phi_i, \gamma_i \rangle$, where $\mathcal{S}_i$ represents the latent state space, $\mathcal{A}_i$ is the action space, $\mathcal{O}_i$ is the observation space, $R_i \colon \mathcal{S}_i \times \mathcal{A}_i \to \mathbb{R}$ is the reward function, $T_i \colon \mathcal{S}_i \times \mathcal{A}_i \to P(\mathcal{S}_i)$ is the transition function, $\phi_i \colon \mathcal{S}_i \times \mathcal{A}_i \to P(\mathcal{O}_i)$ is the observation function, and $\gamma_i$ is the discount factor. By leveraging experiences from previously encountered tasks $\{\mathcal{M}_j\}_{j=1}^{i-1}$, the objective is to adapt the optimal policy $\pi^\star$ that maximizes cumulative rewards to the target task $\mathcal{M}_i$. Here, we consider tasks arriving incrementally in the sequence $\langle \mathcal{M}_1, \ldots, \mathcal{M}_N \rangle$ over time periods $\langle \mathcal{T}_1, \ldots, \mathcal{T}_N \rangle$. In each period $\mathcal{T}_i$, only a replay buffer containing sequences $\{\langle o_t, a_t, r_t \rangle\}_{t=1}^{\mathcal{T}_i}$ from the current task $\mathcal{M}_i$ is available, representing an online setting. While tasks can also be presented offline with predefined source and target tasks, the online framework more closely mirrors human learning, making it a crucial step towards general intelligence.

In this section, we first construct a world model that explicitly embeds the structural relationships among variables in the RL system, and then we show how to encode the changes in the environment by introducing a domain-specific embedding into the model and leveraging it for policy adaptation.

### 2.1 AUGMENTING WORLD MODELS WITH STRUCTURAL RELATIONSHIPS

In POMDPs, extracting latent state representations from high-dimensional observations is crucial for enhancing the efficiency of the decision-making process. World models address this challenge by learning a generative model, which enables agents to predict future states through imagination. These methods typically consider all extracted representations of state variables equally important for policy learning, thereby utilizing all available information regardless of its relevance to the current task. However, real-world tasks often require a focus on specific information. For instance, in the Olympics, swimming speed is crucial in competitive swimming events, but it is less important in synchronized swimming, where grace and precision are prioritized. Hence, it is essential for

agents to understand and focus on task-specific aspects to facilitate effective knowledge transfer by selectively using minimal sufficient information.

To this end, we adopt a causal state representation learning approach that not only enables us to extract state representations, but also to discover structural relationships over the variables. Suppose we observe sequences $\{\langle o_t, a_t, r_t \rangle\}_{t \in \mathcal{T}_i}$ for task $\mathcal{M}_i$, and denote the underlying causal latent states by $s_t \in \mathcal{S}_i$, we formulate the world model into:

$$
\begin{cases}
\text{observation model:} & p_\phi(o_t \mid D^{s \to o} \odot s_t) \\
\text{reward model:} & p_\phi(r_t \mid D^{s \to r} \odot s_t) \\
\text{transition model:} & p_\beta(s_{k,t} \mid D_k^{s \to s} \odot s_{t-1}, D_k^{a \to s} \odot a_{t-1}), \text{ for } k = 1, \ldots, d \\
\text{representation model:} & q_\alpha(s_t \mid s_{t-1}, a_{t-1}, o_t),
\end{cases} \tag{1}
$$

where $s_t = (s_{1,t}, \cdots, s_{d,t})$, $\odot$ is the element-wise product, and $D^{\cdot \to \cdot}$ denote binary masks indicating structural relationships over variables. For instance, if the $j$-th element of $D^{s \to o} \in \{0,1\}^{d \times 1}$ in Eq. (1) is 1, it indicates a causal edge from the state variable $s_{j,t}$ to the current observation signal $o_t$, i.e., $s_{j,t}$ is one of the parents of $o_t$. Consequently, we are supposed to retain $s_{j,t}$ for the observation model. Otherwise, if $D_j^{s \to o} = 0$, then $s_{j,t}$ should be removed from the causal model. Section 3.3 further discusses the estimation procedures for the structural matrices $D$, as well as the corresponding pruning process of the causal model. By learning such causal representations, we can explicitly characterize the decisive factors within each task. However, given that the underlying dynamics often vary across tasks, merely identifying which variables are useful is insufficient. We must also determine how these variables change with the environment for better generalization.

## 2.2 CHARACTERIZATION OF ENVIRONMENTAL CHANGES IN A COMPACT WAY

To address the above need, we now shift our focus to demonstrating how the world model can be modified to ensure robust generalization across the two challenging scenarios, respectively.

**Characterization of Distribution Shifts.** It is widely recognized that changes in the environmental distribution are often caused by modifications in a few specific factors within the data generation process (Ghassami et al., 2018; Schölkopf et al., 2021). In the CoinRun example, such shifts might be due to alterations in background colors ($\varrho$), while other elements remain constant. Therefore, to better characterize these shifts, we introduce a domain-specific change factor, $\theta_i$, that captures the variations across different domains. Concurrently, we leverage $s_t$ to identify the domain-shared latent variables of the environments. This leads us to reformulate Eq. (1) as follows:

$$
\begin{cases}
\text{observation model:} & p_\phi(o_t \mid D^{s \to o} \odot s_t, D^{\theta_i \to o} \odot \theta_i^o) \\
\text{reward model:} & p_\phi(r_t \mid D^{s \to r} \odot s_t, D^{\theta_i \to r} \odot \theta_i^r) \\
\text{transition model:} & p_\beta(s_{k,t} \mid D_k^{s \to s} \odot s_{t-1}, D_k^{\theta_i \to s} \odot \theta_i^s, D_k^{a \to s} \odot a_{t-1}), \text{ for } k = 1, \ldots, d \\
\text{representation model:} & q_\alpha(s_t \mid s_{t-1}, \theta_i, a_{t-1}, o_t),
\end{cases} \tag{2}
$$

where $\theta_i = \{\theta_i^o, \theta_i^r, \theta_i^s\}$ captures essential changes in the observation model, reward model, and transition model, respectively. In CoinRun, this enables us to make quick adaptations by re-estimating $\theta_i^o = \varrho$ in the target task. We assume that the value of $\theta_i$, as well as the structural matrices $D$, remains constant within the same task, but may differ across tasks.

**Characterization of State/Action Space Expansions.** In scenarios where the state or action space expands, we are supposed to add new variables to the existing causal model. The key challenge here is to determine whether the changes stem from distribution shifts or space variations. This dilemma can be addressed using $\theta_i$: If the introduction of $\theta_i$ can well capture the changes in the current observations, it implies that previous tasks $\{\mathcal{M}_j\}_{j=1}^{i-1}$ and $\mathcal{M}_i$ share the same causal variables but exhibit sparse changes in some certain parameters (i.e., distribution shifts). So we only need to store the specific part $\theta_i$ of the causal model for $\mathcal{M}_i$. If it is not the case, then the causal graph must be expanded by adding new causal variables to explain the features unique to $\mathcal{M}_i$.

**Benefits of Explicit Causal Structure.** Upon detecting changes in the environment, we can further leverage the structural constraints $D$ to prune the causal graph. Essentially, we temporarily disregard variables that are irrelevant to the current task. However, for subsequent tasks, we reassess the structural relationships among the variables, enabling potential reuse. This approach allows us to not only preserve previously acquired information but also maintain the flexibility needed to customize the minimal sufficient state variables for each task. Details of this strategy are given in Section 3.

## 2.3 IDENTIFIABLITY OF UNDERLYING WORLD MODELS

In this section, we provide the identifiability theory under different scenarios in this paper: (1) For source task $\mathcal{M}_1$, Theorem 1 establishes the conditions under which the latent variable $s_t$ and the structural matrices $D$ can be identified; (2) For the target task with distribution shifts, Theorem 2 outlines the identifiability of the domain-specific factor $\boldsymbol{\theta}_i$ in linear cases; (3) For the target task with state space shifts, Theorem 3 specifies the identifiability of the newly added state variables $s_t^{\text{add}}$; (4) For the target task that includes both distribution shifts and state space shifts, Corollary 1 demonstrates the identifiability of both $\boldsymbol{\theta}_i$ and $s_t^{\text{add}}$. The proofs are presented in Appendix A. We also discuss the possibility and challenges of establishing the identifiability of $\theta_i^s$ in nonlinear cases in Appendix A.4, followed by empirical results where the learned $\hat{\theta}_i^s$ demonstrates a monotonic correlation with the true values. Below we first introduce the definition of component-wise identifiability, related to Yao et al. (2021), and then we present the theoretical results.

**Definition 1.** *(Component-wise identifiability). Let $\hat{s}_t$ be the estimator of the latent variable $s_t$. Suppose there exists a mapping $h$ such that $s_t = h(\hat{s}_t)$. We say $s_t$ is component-wise identifiable if $h$ is an invertible, component-wise function.*

**Theorem 1.** *(Identifiablity of world model in Eq. (1)). Assume the data generation process in Eq. (3). If the following conditions are satisfied, then $s_t$ is component-wise identifiable: (1) for any $k_1, k_2 \in \{1, \ldots, d\}$ and $k_1 \neq k_2$, $\hat{s}_{k_1,t}$ and $\hat{s}_{k_2,t}$ are conditionally independent given $\hat{s}_{t-1}$; (2) for every possible value of $s_t$, the vector functions defined in Eq. (7) are linearly independent. Furthermore, if the Markov condition and faithfulness assumption hold, then the structural matrices $D$ are also identifiable:*

$$\begin{cases} [o_t, r_{t+1}] &= g(s_t, \epsilon_t) \\ s_t &= g^s(s_{t-1}, a_{t-1}, \epsilon_t^s), \end{cases} \tag{3}$$

*where*

$$\begin{cases} o_t &= g^o(s_t, \epsilon_t^o) \\ r_{t+1} &= g^r(s_t, \epsilon_{t+1}^r). \end{cases} \tag{4}$$

*The $\epsilon_t, \epsilon_t^s, \epsilon_t^o, \epsilon_{t+1}^r$ terms are corresponding independent and identically distributed (i.i.d.) random noises. Following Kong et al. (2023), here we only assume that the global mapping $g$ is invertible.*

**Theorem 2.** *(Identifiability of $\boldsymbol{\theta}_i$ in Eq. (2)). Assume the data generation process in Eq. (5), where the state transitions are linear and additive. If the process encounters distribution shifts and $s_t$ has been identified according to Theorem 1, then $\boldsymbol{\theta}_i$ are component-wise identifiable:*

$$\begin{cases} [o_t, r_{t+1}] &= g(s_t, \theta_i^o, \theta_i^r, \epsilon_t) \\ s_t &= \boldsymbol{A} s_{t-1} + \boldsymbol{B} a_{t-1} + \boldsymbol{C} \theta_i^s + \epsilon_t^s, \end{cases} \tag{5}$$

*where*

$$\begin{cases} o_t &= g^o(s_t, \theta_i^o, \epsilon_t^o) \\ r_{t+1} &= g^r(s_t, \theta_i^r, \epsilon_{t+1}^r). \end{cases} \tag{6}$$

*Following Yao et al. (2021), here we assume that $\boldsymbol{A}$ is full rank, and $\boldsymbol{C}$ is full column rank. We further assume that $s_0 = \hat{s}_0$. Moreover, if the Markov condition and faithfulness assumption hold, the structural matrices $D^{\boldsymbol{\theta}_i \to \cdot}$ are also identifiable.*

**Theorem 3.** *(Identifiability of Expanded State Space). Assume the data generation process in Eq. (3). Consider the expansion of the state space $\mathcal{S}$ by incorporating additional dimensions. Suppose $s_t$ has already been identified according to Theorem 1, then the component-wise identifiability of the newly added variables $s_t^{\text{add}}$ and the additional structural matrices, i.e., $D^{s^{\text{add}} \to \cdot}$ and $D^{\cdot \to s^{\text{add}}}$, can be established if $s_t^{\text{add}}$ (1) represents a differentiable function of $[o_t, r_{t+1}]$, i.e., $s_t^{\text{add}} = f(o_t, r_{t+1})$, and (2) fulfills conditions (1) and (2) specified in Theorem 1.*

**Corollary 1.** *(Identifiability under Multiple Shifts). Assume the data generation process in Eq. (5) involves both distribution shifts and state space shifts that comply with Theorem 2 and Theorem 3, respectively. In this case, both the domain-specific factor $\boldsymbol{\theta}_i$ and the newly added state variable $s_t^{\text{add}}$ are component-wise identifiable.*

## 3 A THREE-STEP SELF-ADAPTIVE STRATEGY FOR MODEL ADAPTATION

In this section, we provide a detailed description of CSR, a strategy aimed at addressing the environmental changes between source and target tasks, thereby ensuring that models can effectively

respond to evolving dynamics. Specifically, we proceed with a three-step strategy: (1) Distribution Shifts Detection and Characterization, (2) State/Action Space Expansions, and (3) Causal Graph Pruning. The overall process of our three-step strategy are described in Fig. 2 and Algorithm 1.

Before this, we first give the estimation procedures for the world models defined in Eq. (2), which follows the state-of-the-art work Dreamer (Hafner et al., 2020; 2023). Given the observations in period $\mathcal{T}_i$, we maximize the objective function $\mathcal{J}^1$, defined as $\mathcal{J} = \mathcal{J}_{\text{rec}} - \mathcal{J}_{\text{KL}} + \mathcal{J}_{\text{reg}}$, for model optimization. The reconstruction part $\mathcal{J}_{\text{rec}}$ is commonly used to minimize the reconstruction error for the perceived observation $o_t$ and the reward $r_t$, which is defined as

$$\mathcal{J}_{\text{rec}} = \mathbb{E}_{q_\alpha} \left( \sum_{t \in \mathcal{T}_i} \{\log p_\phi(o_t \mid D^{\boldsymbol{s} \to o} \odot \boldsymbol{s}_t, D^{\boldsymbol{\theta}_i \to o} \odot \theta_i^o) + \log p_\phi(r_t \mid D^{\boldsymbol{s} \to r} \odot \boldsymbol{s}_t, D^{\boldsymbol{\theta}_i \to r} \odot \theta_i^r)\} \right).$$

We also consider the KL-divergence constraints $\mathcal{J}_{\text{KL}}$ that helps to ensure that the latent representations attain optimal compression of the high-dimensional observations, which is formulated as:

$$\mathcal{J}_{\text{KL}} = \mathbb{E}_{q_\alpha} \left( \sum_{t \in \mathcal{T}_i} \{\lambda_{\text{KL}} \cdot \text{KL}(q_\alpha(s_{k,t} \mid \boldsymbol{s}_{t-1}, \boldsymbol{\theta}_i, a_{t-1}, o_t) \| p_\beta(s_{k,t} \mid D_k^{\boldsymbol{s} \to \boldsymbol{s}} \odot \boldsymbol{s}_{t-1}, D_k^{\boldsymbol{\theta}_i \to \boldsymbol{s}} \odot \theta_i^{\boldsymbol{s}}, D_k^{a \to \boldsymbol{s}} \odot a_{t-1})\} \right),$$

where $\lambda_{\text{KL}}$ is the regularization term. Moreover, as explained below in the Section 3.3, we further use $\mathcal{J}_{\text{reg}}$ as sparsity constraints that help to identify the binary masks $D$ better. Upon implementation, these three components are jointly optimized for model estimation. During the first task $\mathcal{M}_1$, we focus on developing the world models from scratch to capture the compact causal representations effectively. Then, for any subsequent target task $\mathcal{M}_i$ (where $i \geq 2$), our objective shifts to continuously refining the world model to accommodate new tasks according to the following steps.

## 3.1 DISTRIBUTION SHIFTS DETECTION AND CHARACTERIZATION

For each task $\mathcal{M}_i$, our first goal is to determine if it exhibits any distribution shifts. Therefore, in this step, we exclusively updates the domain-specific part $\boldsymbol{\theta}_i$, while keeping all other parameters unchanged from the previous task $\mathcal{M}_{i-1}$, to detect whether the distributions have changed. Recall that the effect of each edge in the structural matrices $D$ can differ from one task to another. By adjusting the values of $\boldsymbol{\theta}_i$, we can also easily characterize the task-specific influence of these connections. Particularly, when $\boldsymbol{\theta}_i$ is set to zero, we temporarily switch the related edges off in task $\mathcal{M}_i$.

Here we adopt forward prediction error (Guo et al., 2020) as the criteria to determine whether the re-estimated model well explains the observations in current task, defined as $\mathcal{L}_{\text{pred}} = \mathbb{E}_{\hat{o}_{t+1} \sim p_\phi} \| \hat{o}_{t+1} - o_{t+1} \|_2^2$. A corresponding threshold, $\tau^\star$, is established. Upon implementation, we use the final prediction loss of the model on the source task $\mathcal{M}_1$ as the threshold value, thereby avoiding the need for manual setting. If the model's performance $\tau$ is below this expected threshold, it means that the current task $\mathcal{M}_i$ shares the same causal variables with previous tasks, requiring only sparse changes of some parameters in the world model, and then we only need to re-estimate the specific part $\boldsymbol{\theta}_i$ to effectively manages these distribution changes. Otherwise, we proceed to the next step.

## 3.2 STATE/ACTION SPACE EXPANSIONS

When the involved domain-specific features $\boldsymbol{\theta}_i$ fail to accurately represent the target task $\mathcal{M}_i$, it becomes essential to incorporate additional causal variables into the existing causal graph to account for the features encountered in the new task. Given that the action variables are observable, we can directly obtain the relevant information when the action space expands. Thus, in this step, we focus on developing strategies to effectively manage state expansions. Let $d'$ denote the number of causal variables to be added. We first determine the value of $d'$ and introduce new causal features. Following this decision stage, we extend the causal representations from $\boldsymbol{s}_t$ to $\boldsymbol{s}'_t = (\boldsymbol{s}_t, \boldsymbol{s}_t^{\text{add}})$, where $\boldsymbol{s}_t^{\text{add}} = (s_{d+1,t}, \dots, s_{d+d',t})$, by incorporating the additional $d'$ causal variables. This is implemented by increasing the dimensions of input/output layers of the world models. For instance, the state input of the transition model will increase from $d$ to $d + d'$. Accordingly, we focus on learning the newly incorporated components with only a few samples, as the previous model has already captured the existing relationships between variables. This approach allows us to leverage prior knowledge effectively and achieve low-cost knowledge transfer. Specifically, we propose the following three implementations for state space expansion:

---

[1]Formally written as $\mathcal{J}(\phi, \beta, \alpha, \boldsymbol{\theta}_i, D)$; arguments are omitted for brevity.

1. *Random (Rnd):* $d'$ is randomly sampled from a uniform distribution.

2. *Deterministic (Det):* It sets a constant value for $d'$. However, this approach may overlook task-specific differences, potentially leading to either insufficient or redundant expansions. To address this, we employ group sparsity regularization (Yoon et al., 2017) on the added parameters after deterministic expansion, which allows for expansion while retaining the capability of shrinking.

3. *Self-Adaptive (SA):* It searches for the value of $d'$ that best fits the current task. To achieve this, we transform expansion into a decision-making process by considering the number of causal variables added to the graph as actions. Inspired by Xu & Zhu (2018), we define the state variable to reflect the current causal graph and derive the reward based on changes in predictive accuracy, which is calculated as the differences of the model's prediction errors before and after expansion. Details are given in Appendix B.

It is noteworthy that our method allows for flexibility in the choice of expansion strategies. Intuitively, in our case, the Self-Adaptive approach is most likely to outperform others, and the experimental results in Section 4 further verify this point.

### 3.3 CAUSAL GRAPH PRUNING

As discussed in Section 2.1, not all variables contribute significantly to policy learning, and the necessary subset also differs between tasks. Therefore, during the generalization process, it is essential to identify the minimal sufficient state variables for each task. Fortunately, with the estimated causal model that explicitly encodes the structural relationships between variables, we can categorize these state variables $s_t$ into the following two classes:

1. *Compact state representation $s_{k,t}^c$:* A variable that either affects the observation $o_t$, or the reward $r_{t+1}$, or influences other state variables $s_{j,t+1}$ ($k \neq j$) at the next time step (i.e., $D^{s \to o} = 1$, or $D^{s \to r} = 1$, or $D_{j,k}^{s \to s} = 1$, e.g., $s_{1,t}$ in Fig. 2).

2. *Non-compact state representation $s_{k,t}^{\bar{c}}$:* A variable that does not meet the criteria for a compact state representation (e.g., $s_{2,t}$ in Fig. 2).

Similarly, the change factors $\theta_i$ can be classified in the same manner. These definitions allow us to selectively remove non-compact ones, thereby pruning the causal graph. That is, a variable is retained only when its corresponding structural constraints $D$ are non-zero. Hence, to better characterize the binary masks $D$ and the sparsity of $\theta_i$, we define a regularization term $\mathcal{J}_{\text{reg}}$ by leveraging the edge-minimality property (Zhang & Spirtes, 2011), formulated as

$$\mathcal{J}_{\text{reg}} = -\lambda_{\text{reg}} \left[ \|D^{s \to o}\|_1 + \|D^{s \to r}\|_1 + \|D^{s \to s}\|_1 + \|D^{a \to s}\|_1 + \|D^{\theta_i \to s}\|_1 + \|\theta_i\|_1 \right],$$

where $\lambda_{\text{reg}}$ represents the regularization term. Incorporating the regularization term $\mathcal{J}_{\text{reg}}$ directly into the objective function $\mathcal{J}$ confers a notable advantage: it enables concurrent pruning and model training. Specifically, the presence of $\mathcal{J}_{\text{reg}}$ induces certain entries of $D$ to transition from 1 to 0 during model estimation, thereby promoting sparsity naturally without additional training phases.

### 3.4 LOW-COST POLICY GENERALIZATION UNDER DIFFERENT SCENARIOS

After identifying what and where the changes occur, we are now prepared to perform policy generalization to the target task $\mathcal{M}_i$. Given that the number of state variables varies between the distribution shifts and state space expansion scenarios, the strategy for policy transfer also differs.

According to above definitions, in all these tasks, we incorporate both the domain-shared state representation $s_t^c$ and the domain-specific change factor $\theta_i^c$ as inputs to the policy $\pi^\star$, represented as: $a_t = \pi^\star(s_t^c, \theta_i^c)$. This approach enables the agent to accommodate potentially variable aspects of the environment during policy learning. Consequently, given the re-estimated value of the compact state representation $s_t^c$ and the compact change factor $\theta_i^c$ for task $\mathcal{M}_i$, if the model's prediction error in the Distribution Shifts Detection and Characterization step meets expectations, we can directly transfer the learned policy $\mathcal{M}_i$ by applying $a_t = \pi^\star(s_t^c, \theta_i^c)$.

Otherwise, if the state variables expand from $s_t^c$ to $s_t^{c\prime}$, along with the updated change factor $\theta_i^{c\prime}$, we then relearn the policy $\pi^{\star\prime}$ basing on $\pi^\star$. Similarly, we train the newly added structures in the policy network while finetuning the original parameters, thereby updating the policy to $a_t = \pi^{\star\prime}(s_t^{c\prime}, \theta_i^{c\prime})$.

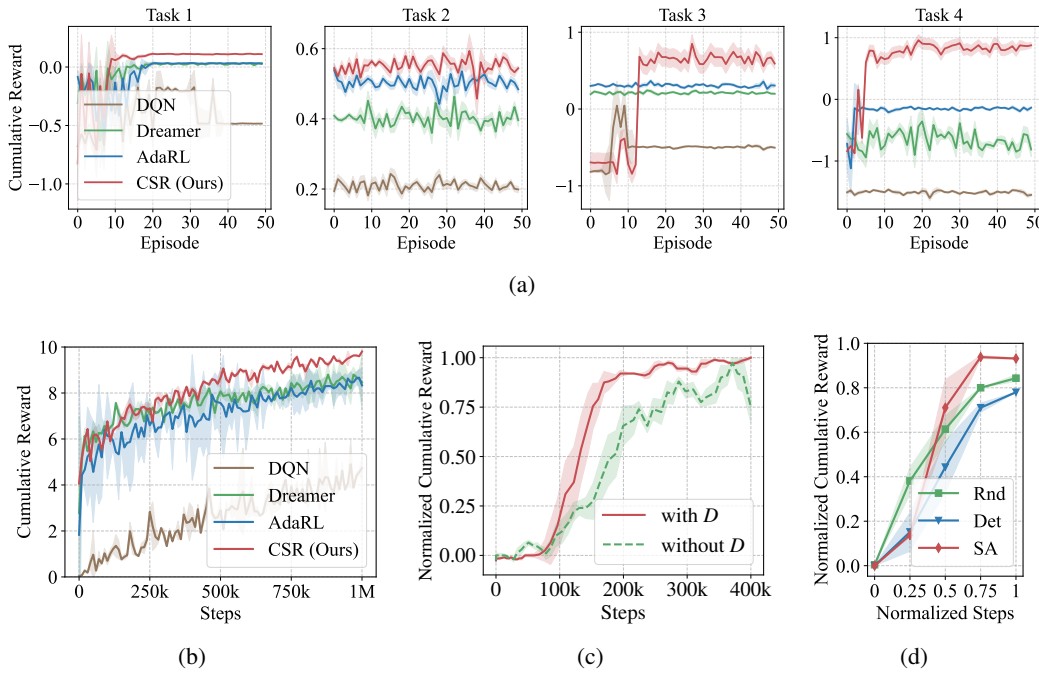

Figure 3: **Experimental results that answer the key questions in Section 4: (Q1)** CSR demonstrates the best generalization capability compared to baseline methods in (a) Simulation and (b) CoinRun; **(Q2)** CSR with structural embeddings $D$ significantly outperforms CSR without $D$ in Atari games; **(Q3)** The SA expansion strategy yields the highest normalized average training episodic return in our experiments.

## 4 EXPERIMENTS

We evaluate the generalization capability of CSR on a number of simulated and well-established datasets[2], including the CartPole, CoinRun and Atari environments, with detailed descriptions provided in Appendix D.3. For all these benchmarks, we evaluate the POMDP case, where the inputs are high-dimensional observations. Specifically, the evaluation focuses on answering the following key questions:

- **Q1:** Can CSR effectively detect and adapt to the two types of environmental changes?
- **Q2:** Does the incorporation of causal knowledge enhance the generalization performance?
- **Q3:** Is searching for the optimal expansion structure necessary?

We compare our approach against several baselines: Dreamer (Hafner et al., 2023), which handles fixed tasks without integrating causal knowledge; AdaRL (Huang et al., 2021), which employs simple scenario-based policy adaptation without space expansion considerations; and the traditional model-free DQN (Mnih et al., 2015) and SPR (Schwarzer et al., 2020). Additionally, for the Atari games, we benchmark against the state-of-the-art method, EfficientZero (Ye et al., 2021). All results are averaged over 5 runs, more implementation details can be found in Appendix D.

**CSR consistently exhibits the best adaptation capability across all these environments (Q1). Simulated Experiments.** In our simulated experiments, we conducted a sequence of four tasks following the procedures outlined in Appendix D.3.1. To evaluate the performance of different methods across varying scenarios, Task 2 focuses exclusively on distribution shifts, Task 3 addresses changes solely within the environment space, and Task 4 combines both distribution and space changes. As shown in Fig. 3(a), CSR consistently outperforms the baselines in adapting to new tasks, particularly excelling in scenarios with space variations. This demonstrates CSR's ability to accurately detect and adjust to the changes in the environment. Moreover, CSR tends to converge faster toward higher rewards, underscoring its efficiency in data utilization during generalization.

---

[2]Code is available at https://github.com/CMACH508/CSR.

| Model | Scores | | | | Minimum Adaptation Steps | | | |
|---|---|---|---|---|---|---|---|---|
| | Task 1 | Task 2 | Task 3 | Task 4 | Task 1 | Task 2 | Task 3 | Task 4 |
| DQN | 102.8 ($\pm$ 7.8) | 65.6 ($\pm$ 15.6) | 107.4 ($\pm$ 17.0) | 104.0 ($\pm$ 12.0) | ✗ | ✗ | ✗ | ✗ |
| Dreamer | 500.0 ($\pm$ 0.0) | 397.6 ($\pm$ 7.2) | 311.6 ($\pm$ 23.3) | 356.3 ($\pm$ 59.9) | 50k | ✗ | ✗ | ✗ |
| AdaRL | 500.0 ($\pm$ 0.0) | 468.0 ($\pm$ 2.3) | 410.0 ($\pm$ 3.4) | 407.5 ($\pm$ 34.5) | 50k | 4k | ✗ | ✗ |
| CSR (ours) | **500.0** ($\pm$ 0.0) | **500.0** ($\pm$ 0.0) | **500.0** ($\pm$ 0.0) | **500.0** ($\pm$ 0.0) | **50k** | **2k** | **4k** | **10k** |

Table 1: **Only CSR consistently adapts to successive environmental changes in CartPole.** Evaluation results of various approaches are presented, with a maximum episode length of 500. 'Minimum Adaptation Steps' refers to the minimal amount of data required for model generalization, as illustrated in Fig. 8. ✗ denotes that a method fails to adapt under limited training steps due to non-convergence or suboptimal performance.

| Task | Random | SPR | Dreamer | AdaRL | EfficientZero | CSR (ours) |
|---|---|---|---|---|---|---|
| Alien | 291.9 ($\pm$ 83.5) | 970.3 ($\pm$ 311.1) | 1010.1 ($\pm$ 339.0) | 1147.5 ($\pm$ 125.7) | 557.4 ($\pm$ 185.9) | **1586.9** ($\pm$ 127.0) |
| Bank Heist | 18.4 ($\pm$ 2.5) | 110.1 ($\pm$ 128.2) | 1313.7 ($\pm$ 341.5) | 1285.4 ($\pm$ 131.8) | 181.0 ($\pm$ 90.6) | **1454.1** ($\pm$ 178.8) |
| Crazy Climber | 9668.0 ($\pm$ 2286.0) | 32723.5 ($\pm$ 12125.2) | 68026.5 ($\pm$ 15628.6) | 62565.3 ($\pm$ 15162.2) | 56408.3 ($\pm$ 13388.0) | **88306.5** ($\pm$ 18029.6) |
| Gopher | 235.6 ($\pm$ 42.5) | 294.0 ($\pm$ 312.1) | 5607.3 ($\pm$ 1982.9) | 5359.6 ($\pm$ 1736.2) | 1083.2 ($\pm$ 784.8) | **6718.6** ($\pm$ 1703.1) |
| Pong | -20.2 ($\pm$ 0.1) | -6.8 ($\pm$ 14.3) | 18.0 ($\pm$ 3.1) | 17.6 ($\pm$ 2.7) | 6.8 ($\pm$ 7.3) | **19.6** ($\pm$ 1.1) |

Table 2: **CSR outperforms every baseline method on the selected set of Atari games.**

**CartPole Experiments.** CartPole is a classic control task where players move a cart left or right to balance a pendulum attached on it. In our experiments, we consider four consecutive tasks. Task 2 focuses exclusively on distribution shifts by randomly selecting the cart mass and the gravity from $\{0.5, 1, 2.5, 3.5, 4.5\}$ and $\{5, 9.8, 20, 30, 40\}$, respectively. In Tasks 1 and 2, we disregard the influence of the friction force between the cart and the track. In Task 3, however, we introduce this friction into the environment, and vary it over time, which simulates a game scenario where the cart moves on different surfaces, such as ice or grass (see Fig. 6 and Fig. 7). We reflect these changes in the observations by visualizing the track with different colored segments. To explore the generalization capabilities of the proposed method in scenarios with action expansion, we further designed Task 4, in which we expand the action space by incorporating additional possible force values that can be applied to the cart. The evaluation outcomes for these models are summarized in Table 1. We find that CSR consistently achieves the highest scores across all tasks, demonstrating its capability to promptly detect and adapt to environmental changes. In contrast, other baseline methods struggle to adjust to the introduction of the new friction variable and actions.

**CoinRun Experiments.** The learning curves in Fig. 3(b) depict our method's consistent superiority over the baselines during knowledge transferring from low to high difficulty levels in CoinRun. We also observe that model-based methods tend to generalize more quickly than model-free ones in our experiments. This finding suggests that the forward-planning capabilities of world models confer significant advantages in adaptation. Visualizations of the reconstructed observations from various methods are presented in Appendix D.3.3.

**Atari Experiments.** We also conduct a series of interesting experiments on the Atari 100K games, which includes 26 games with a budget of 400K environment steps (Kaiser et al., 2019). Specifically, we select five representative games for evaluation: Alien, Bank Heist, Carzy Climber, Gopher, and Pong. The modes and difficulties available in each game are summarized in Table 6. For each of these games, we perform experiments among a sequence of four tasks, where each task randomly assigns a (mode, difficulty) pair. We then train these models on the source task and generalize them to downstream target tasks. Table 2 summarizes the average final scores across these tasks. We see that CSR achieves the highest mean scores in all the five games. Moreover, Fig. 16 illustrates the average generalization performance of various methods on downstream target tasks, while Fig. 17

to Fig. 21 present the training curves for each game, respectively. The reconstructions, as well as the estimated structural matrices, are provided in Appendix D.3.4.

**Integrating causal knowledge by explicitly embedding structural matrices $D$ into the world model improves the generalization ability (Q2).** Figure 3(c) illustrates the average performance of CSR with and without $D$ in Atari games. We observe a significantly faster and higher increase in the cumulative reward when taking structural relationships into consideration. This demonstrates the efficiency enhancement in policy learning through the removal of redundant causal variables, which accelerates the extraction and utilization of knowledge during the generalization process.

**Searching for the optimal expansion structure brings notable performance gains but involves a trade-off (Q3).** We conduct comparative experiments using the three methods described in Section 3.2 for all the environments and average them into Fig. 3(d). The results demonstrate that seeking for the optimal structure significantly improves expansion performance, leading us to apply the *Self-Adaptive* approach. However, we also observe that each search step requires extensive training time for models with different expansion scales, making the search process highly time-consuming. Therefore, it is crucial to consider this trade-off in practical applications.

## 5 RELATED WORK

Recently, extensive research efforts have been invested in learning abstract representations in RL, employing methodologies such as image reconstruction (Watter et al., 2015), contrastive learning (Sermanet et al., 2018; Mazoure et al., 2020), and the development of world models (Sekar et al., 2020). A prominent research avenue within this domain is causal representation learning, which aims to identify high-level causal variables from low-level observations, thereby enhancing the accuracy of information available for decision-making processes (Schölkopf et al., 2021). Approaches such as ASRs (Huang et al., 2022) and IFactor (Liu et al., 2023) leverage causal factorization and structural constraints within causal variables to develop more accurate world models. Moreover, CDL (Wang et al., 2022), GRADER (Ding et al., 2022) and Causal Exploration (Yang et al., 2024) seek to boost exploration efficiency by learning causal models. Despite these advancements, many of these studies are tailored to specific tasks and struggle to achieve the level of generalization across tasks where human performance is notably superior (Taylor & Stone, 2009; Zhou et al., 2023).

To overcome these limitations, Harutyunyan et al. (2015) develop a reward-shaping function that captures the target task's information to guide policy learning. Taylor et al. (2007) and Zhang et al. (2020) aim to map tasks to invariant state variables, thereby learning policies robust to environmental changes. AdaRL (Huang et al., 2021) is dedicated to learning domain-shared and domain-specific representations to facilitate policy transfer. Distinct from these works, CSP (Gaya et al., 2022) approaches from the perspective of policy learning directly, by incrementally constructing a subspace of policies to train agents. However, most of these works assume a constant environment space, which is often not the case in practical applications. Therefore, in this paper, we investigate the feasibility of knowledge transfer when the state space can also change. Furthermore, the approach we propose is also related to the area of dynamic neural networks, where various methods have been developed to address sequences of tasks that require dynamical modifications to the network architecture, such as DEN (Yoon et al., 2017), PackNet (Mallya & Lazebnik, 2018), APD (Yoon et al., 2019), CPG (Hung et al., 2019), and Learn-to-Grow (Li et al., 2019).

## 6 CONCLUSIONS AND FUTURE WORK

In this paper, we explore a broader range of scenarios for generalizable reinforcement learning, where changes across domains arise not only from distribution shifts but also space expansions. To investigate the adaptability of RL methods in these challenging scenarios, we introduce CSR, an approach that uses a three-step strategy to enable agents to detect environmental changes and autonomously adjust as needed. Empirical results from various complex environments, such as CartPole, CoinRun and Atari games, demonstrate the effectiveness of CSR in generalizing across evolving tasks. The primary limitation of this work is that it only considers generalization across domains and does not account for nonstationary changes over time. Therefore, a future research direction is to develop methods to automatically detect and characterize nonstationary changes both over time and across tasks.

## ACKNOWLEDGMENTS

Yupei Yang, Shikui Tu and Lei Xu would like to acknowledge the support by the Shanghai Municipal Science and Technology Major Project, China (Grant No. 2021SHZDZX0102), and by the National Natural Science Foundation of China (62172273).

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

# A  PROOFS OF IDENTIFIABLITY THEORY

Before presenting the proofs, we first introduce the relevant notations and assumptions.

**Notations.**   We denote the underlying state variable by $s_t = \{s_{1,t}, \ldots, s_{d,t}\}$ and denote $o_t$ as the observation. Also, we denote the mapping from state estimator $\hat{s}_t$ to state $s_t$ by $s_t = h(\hat{s}_t)$, and denote the Jacobian matrix of $h$ as $\mathbf{J}_t^h$. Let $\zeta_{k,t} \triangleq \log p(s_{k,t} \mid s_{t-1})$, we further denote

$$\boldsymbol{\omega}_{k,t} \triangleq \left( \frac{\partial^2 \zeta_{k,t}}{\partial s_{k,t} \partial s_{1,t-1}}, \frac{\partial^2 \zeta_{k,t}}{\partial s_{k,t} \partial s_{2,t-1}}, \ldots, \frac{\partial^2 \zeta_{k,t}}{\partial s_{k,t} \partial s_{d,t-1}} \right)^\top, \mathring{\boldsymbol{\omega}}_{k,t} \triangleq \left( \frac{\partial^3 \zeta_{k,t}}{\partial s_{k,t}^2 \partial s_{1,t-1}}, \frac{\partial^3 \zeta_{k,t}}{\partial s_{k,t}^2 \partial s_{2,t-1}}, \ldots, \frac{\partial^3 \zeta_{k,t}}{\partial s_{k,t}^2 \partial s_{d,t-1}} \right)^\top. \tag{7}$$

**Assumption 1.** *$\zeta_{k,t}$ is twice differentiable with respect to $s_{k,t}$ and differentiable with respect to $s_{l,t-1}$, for all $l \in \{1, \ldots, d\}$.*

**Assumption 2** (Faithfulness assumption). *For a causal graph $\mathcal{G}$ and the associated probability distribution $P$, every true conditional independence relation in $P$ is entailed by the Causal Markov Condition applied to $\mathcal{G}$ (Spirtes et al., 2001).*

## A.1  PROOF OF THEOREM 1

Based on aforementioned assumptions and definitions, Theorem 1 establishes the conditions for the component-wise identifiablity of the state variable $s_t$ and the structural matrices $D$ in Eq. (1).

**Theorem 1.** *(Identifiablity of world model in Eq. (1)). Assume the data generation process in Eq. (8). If the following conditions are satisfied, then $s_t$ is component-wise identifiable: (1) for any $k_1, k_2 \in \{1, \ldots, d\}$ and $k_1 \neq k_2$, $\hat{s}_{k_1,t}$ and $\hat{s}_{k_2,t}$ are conditionally independent given $\hat{s}_{t-1}$; (2) for every possible value of $s_t$, the vector functions defined in Eq. (7) are linearly independent. Furthermore, if the Markov condition and faithfulness assumption hold, then the structural matrices $D$ are also identifiable:*

$$\begin{cases} [o_t, r_{t+1}] &= g(s_t, \epsilon_t) \\ s_t &= g^{\boldsymbol{s}}(s_{t-1}, a_{t-1}, \epsilon_t^{\boldsymbol{s}}), \end{cases} \tag{8}$$

*where*

$$\begin{cases} o_t &= g^o(s_t, \epsilon_t^o) \\ r_{t+1} &= g^r(s_t, \epsilon_{t+1}^r). \end{cases} \tag{9}$$

*The $\epsilon_t, \epsilon_t^{\boldsymbol{s}}, \epsilon_t^o, \epsilon_{t+1}^r$ terms are corresponding independent and identically distributed (i.i.d.) random noises. Following Kong et al. (2023), here we only assume that the global mapping $g$ is invertible.*

*Proof.* The proof proceeds in two steps. First, we demonstrate that the data generation process in Eq. (8) is equivalent to the noiseless data distribution. Second, we summarize the proof steps of the identifiablity of the state variables $s_t$ under the noiseless distribution, which has already been provided in Yao et al. (2022).

**Step 1: transform into noise-free distributions.**

Let $y_t = [o_t, r_{t+1}]$. We denote $p(y_t) = \int p_g(y_t|s_t)p_\kappa(s_t)ds_t$ where $g, \kappa$ are the parameters of the probability functions. Suppose $p_{g,\kappa}(y_t) = p_{\hat{g},\hat{\kappa}}(y_t)$ holds for all $y_t$, where $(g, \kappa)$ and $(\hat{g}, \hat{\kappa})$ are two sets of parameters. We complete the proof primarily by following Khemakhem et al. (2020).

By applying the law of total probability, we have

$$\int_{\mathcal{S}} p_\kappa(s_t) \cdot p_g(y_t|s_t)ds_t = \int_{\mathcal{S}} p_{\hat{\kappa}}(s_t) \cdot p_{\hat{g}}(y_t|s_t)ds_t. \tag{10}$$

Further define $p_g(y_t|s_t) = p_{\epsilon_t}(y_t - g(s_t))$, we get

$$\int_{\mathcal{S}} p_\kappa(s_t) \cdot p_{\epsilon_t}(y_t - g(s_t))ds_t = \int_{\mathcal{S}} p_{\hat{\kappa}}(s_t) \cdot p_{\epsilon_t}(y_t - \hat{g}(s_t))ds_t. \tag{11}$$

Replacing $\overline{y}_t = g(s_t)$ on the left hand side, and similarly on the right hand side, we obtain

$$\int_{\mathcal{O}} p_\kappa(g^{-1}(\overline{y}_t)) \text{ vol } \mathbf{J}_t^{g^{-1}}(\overline{y}_t) \cdot p_{\epsilon_t}(y_t - \overline{y}_t)d\overline{y}_t = \int_{\mathcal{O}} p_{\hat{\kappa}}(\hat{g}^{-1}(\overline{y}_t)) \text{ vol } \mathbf{J}_t^{\hat{g}^{-1}}(\overline{y}_t) \cdot p_{\epsilon_t}(y_t - \overline{y}_t)d\overline{y}_t, \tag{12}$$

where vol $\mathbf{J} = \sqrt{\det \mathbf{J}^\top \mathbf{J}}$.

By introducing $\tilde{p}_{g,\kappa}(\overline{y}_t) = p_\kappa(g^{-1}(\overline{y}_t)) \text{ vol } \mathbf{J}_t^{g^{-1}}(\overline{y}_t) \mathbb{1}(\overline{y}_t)$ on both sides, we can rewrite Eq. (12) as

$$\int_{\mathbb{R}^\upsilon} \tilde{p}_{g,\kappa}(\overline{y}_t) \cdot p_{\epsilon_t}(y_t - \overline{y}_t)d\overline{y}_t = \int_{\mathbb{R}^\upsilon} \tilde{p}_{\hat{g},\hat{\kappa}}(\overline{y}_t) \cdot p_{\epsilon_t}(y_t - \overline{y}_t)d\overline{y}_t, \tag{13}$$

where $\upsilon = \dim \mathcal{O} + \dim R$. By the definition of convolution, Eq. (13) is equivalent to

$$(\tilde{p}_{g,\kappa} * p_\epsilon)(y_t) = (\tilde{p}_{\hat{g},\hat{\kappa}} * p_\epsilon)(y_t), \tag{14}$$

where $*$ denote the convolution operator. Denote $F[.]$ the Fourier transform and $\varphi_\epsilon = F[p_\epsilon]$, we have

$$F[\tilde{p}_{g,\kappa}](\Omega)\varphi_\epsilon(\Omega) = F[\tilde{p}_{\hat{g},\hat{\kappa}}](\Omega)\varphi_\epsilon(\Omega). \tag{15}$$

Assume set $\{\boldsymbol{x} \in \mathcal{X} | \varphi_\epsilon(\boldsymbol{x}) = 0\}$ has measure zero, we can drop $\varphi_\epsilon(\Omega)$ from both sides, which obtains

$$F[\tilde{p}_{g,\kappa}](\Omega) = F[\tilde{p}_{\hat{g},\hat{\kappa}}](\Omega). \tag{16}$$

Therefore, for all $y_t \in \mathcal{O} \times R$, we have

$$\tilde{p}_{g,\kappa}(y_t) = \tilde{p}_{\hat{g},\hat{\kappa}}(y_t). \tag{17}$$

This indicates that the noise-free distributions must coincide for the overall distributions to remain identical after adding noise, effectively reducing the noisy case in Eq. (8) into the noiseless case.

**Step 2: establish identifiability of state variables.**

After transforming the problem into the noise-free case, we proceed by summarizing the key proof steps of the identifiability of the state variables $\boldsymbol{s}_t$, following Yao et al. (2022), which are:

- First, by making use of the conditional independence of the components of $\hat{\boldsymbol{s}}_t$ given $\hat{\boldsymbol{s}}_{t-1}$, it is shown that:

$$\frac{\partial^2 \log p(\hat{\mathbf{s}}_t \mid \hat{\mathbf{s}}_{t-1})}{\partial \hat{s}_{i,t} \partial \hat{s}_{j,t}} = 0. \tag{18}$$

- Second, by utilizing the Jacobian matrix $\mathbf{J}_t^h$ to calculate Eq. (18), it is derived that

$$\frac{\partial^3 \log p\left(\hat{\boldsymbol{s}}_t \mid \hat{\boldsymbol{s}}_{t-1}\right)}{\partial \hat{s}_{i,t} \partial \hat{s}_{j,t} \partial s_{l,t-1}} = \sum_{k=1}^d \left( \frac{\partial^3 \zeta_{k,t}}{\partial s_{k,t}^2 \partial s_{l,t-1}} \cdot \mathbf{J}_{k,i,t}^h \mathbf{J}_{k,j,t}^h + \frac{\partial^2 \zeta_{k,t}}{\partial s_{k,t} \partial s_{l,t-1}} \cdot \frac{\partial \mathbf{J}_{k,i,t}^h}{\partial \hat{s}_{j,t}} \right) \equiv 0. \tag{19}$$

- Finally, it is established that $\boldsymbol{s}_t$ is identifiable, up to an invertible, component-wise nonlinear transformation of a permuted version of $\hat{\boldsymbol{s}}_t$, if the linear independence of vector funtions defined in Eq. (7) holds and the the Jacobian matrix $\mathbf{J}_t^h$ satisfies Eq. (19).

Moreover, the proofs for the identifiablity of the structural matrices $D$ are presented in Huang et al. (2021). Based on these steps, we next provide the proofs of Theorem 2-3, and Corollary 1. $\qquad \square$

## A.2 PROOF OF THEOREM 2

Different from existing methods, to capture the changing dynamics in the environment, we have introduced a task-specific change factor, $\boldsymbol{\theta}_i$, into the world model as defined in Eq. (2). Accordingly, Theorem 2 presents the identifiability of $\boldsymbol{\theta}_i$ and the corresponding structural matrices $D^{\boldsymbol{\theta}_i \to \cdot}$ for scenarios involving only distribution shifts in linear cases.

**Theorem 2.** *(Identifiability of $\boldsymbol{\theta}_i$ in Eq. (2)). Assume the data generation process in Eq. (20), where the state transitions are linear and additive. If the process encounters distribution shifts and $\boldsymbol{s}_t$ has been identified according to Theorem 1, then $\boldsymbol{\theta}_i$ are component-wise identifiable:*

$$\begin{cases} [o_t, r_{t+1}] & = & g(\boldsymbol{s}_t, \theta_i^o, \theta_i^r, \epsilon_t) \\ \boldsymbol{s}_t & = & \boldsymbol{A}\boldsymbol{s}_{t-1} + \boldsymbol{B}a_{t-1} + \boldsymbol{C}\theta_i^s + \epsilon_t^s, \end{cases} \tag{20}$$

*where*

$$\begin{cases} o_t & = & g^o(\boldsymbol{s}_t, \theta_i^o, \epsilon_t^o) \\ r_{t+1} & = & g^r(\boldsymbol{s}_t, \theta_i^r, \epsilon_{t+1}^r). \end{cases} \tag{21}$$

*Following Yao et al. (2021), here we assume that $\boldsymbol{A}$ is full rank, and $\boldsymbol{C}$ is full column rank. We further assume that $\boldsymbol{s}_0 = \hat{\boldsymbol{s}}_0$. Moreover, if the Markov condition and faithfulness assumption hold, the structural matrices $D^{\boldsymbol{\theta}_i \to \cdot}$ are also identifiable.*

*Proof.* The proofs for the identifiablity of $D^{\boldsymbol{\theta}_i \to \cdot}$ are given in Huang et al. (2021). Here, we provide the proof for the identifiablity of $\boldsymbol{\theta}_i$, which is done in the following four steps:

- In step 1, we prove that $\theta_i^o$ can be identified up to component-wise transformation when only the observation function exhibits distribution shifts.

- In step 2, we demonstrate that $\theta_i^r$ is component-wise identifiable when only the reward function experiences distribution shifts.

- In step 3, we show that $\theta_i^s$ can be identified component-wisely when only the transition function undergoes distribution shifts.

- In step 4, we establish that in the general case, where the observation, reward, and transition functions may undergo distribution shifts simultaneously, $\boldsymbol{\theta}_i = \{\theta_i^o, \theta_i^r, \theta_i^s\}$ is identifiable.

**Step 1: prove the identifiability of $\theta_i^o$.**

According to Eq. (21), we have
$$y_t = g(\boldsymbol{s}_t, \theta_i^o, \epsilon_t). \tag{22}$$

Denote $\boldsymbol{x}_t = (\boldsymbol{s}_t, \theta_i^o)$. There exists
$$\boldsymbol{x}_t = h'(\hat{\boldsymbol{x}}_t), \tag{23}$$
where $h' = g^{-1} \circ \hat{g}$, and $\hat{\boldsymbol{x}}_t$ is the estimator of $\boldsymbol{x}_t$. Since both $g$ and $\hat{g}$ are invertible, $h'$ is invertible. Therefore, we have
$$\mathbf{J}_t^{h'} = \begin{bmatrix} \frac{\partial \boldsymbol{s}_t}{\partial \hat{\boldsymbol{s}}_t} & \frac{\partial \boldsymbol{s}_t}{\partial \hat{\theta}_i^o} \\ \frac{\partial \theta_i^o}{\partial \hat{\boldsymbol{s}}_t} & \frac{\partial \theta_i^b}{\partial \hat{\theta}_i^o} \end{bmatrix}, \tag{24}$$

where $\mathbf{J}_t^{h'}$ is full rank. Note that $\frac{\partial \boldsymbol{s}_t}{\partial \hat{\theta}_i^o} = 0$ and $\frac{\partial \theta_i^o}{\partial \hat{\boldsymbol{s}}_t} = 0$. Further recall that we assume the identifiability of $\boldsymbol{s}_t$, which means that $\mathbf{J}_t^h = \frac{\partial \boldsymbol{s}_t}{\partial \hat{\boldsymbol{s}}_t}$ is full rank. We can derive that $\frac{\partial \theta_i^o}{\partial \hat{\theta}_i^o}$ must be full rank. That is, $\theta_i^o$ is component-wise identifiable.

**Step 2: prove the identifiability of $\theta_i^r$.**

If only the reward function $g^r$ exhibits distribution shifts, we have
$$y_t = g(\boldsymbol{s}_t, \theta_i^r, \epsilon_t). \tag{25}$$

It is straightforward to see that $\theta_i^r$ is blockwise identifiable using the same technique in Step 1.

**Step 3: prove the identifiability of $\theta_i^s$.**

Recall that we have
$$\boldsymbol{s}_t = \boldsymbol{A}\boldsymbol{s}_{t-1} + \boldsymbol{B}a_{t-1} + \boldsymbol{C}\theta_i^s + \epsilon_t^s. \tag{26}$$

By leveraging the recursive property of the state transition process, we can derive that
$$\boldsymbol{s}_t = \boldsymbol{A}^t \boldsymbol{s}_0 + \left(\sum_{k=0}^{t-1} \boldsymbol{A}^k \boldsymbol{B}\right) a_{t-1-k} + \left(\sum_{k=0}^{t-1} \boldsymbol{A}^k\right) \boldsymbol{C}\theta_i^s + \left(\sum_{k=0}^{t-1} \boldsymbol{A}^k\right) \epsilon_{t-k}. \tag{27}$$

Similarly for $\hat{\boldsymbol{s}}_t$, we have
$$\hat{\boldsymbol{s}}_t = \hat{\boldsymbol{A}}^t \hat{\boldsymbol{s}}_0 + \left(\sum_{k=0}^{t-1} \hat{\boldsymbol{A}}^k \hat{\boldsymbol{B}}\right) a_{t-1-k} + \left(\sum_{k=0}^{t-1} \hat{\boldsymbol{A}}^k\right) \hat{\boldsymbol{C}}\theta_i^s + \left(\sum_{k=0}^{t-1} \hat{\boldsymbol{A}}^k\right) \epsilon_{t-k}. \tag{28}$$

Note that $\boldsymbol{s}_0 = \hat{\boldsymbol{s}}_0$. Therefore, combining Eq. (27) and Eq. (28) gives
$$\left(\sum_{k=0}^{t-1} \boldsymbol{A}^k\right) \boldsymbol{C}\theta_i^s = \boldsymbol{s}_t - \boldsymbol{A}^t \hat{\boldsymbol{A}}^{-t} \left[\hat{\boldsymbol{s}}_t - \left(\sum_{k=0}^{t-1} \hat{\boldsymbol{A}}^k\right) \hat{\boldsymbol{C}}\theta_i^s\right] + \Theta, \tag{29}$$

where $\Theta$ is a constant term. Taking the derivative w.r.t $\hat{\theta}_i^s$ on both sides, we obtain

$$M \frac{\partial \theta_i^s}{\partial \hat{\theta}_i^s} = \frac{\partial s_t}{\partial \hat{s}_t} \frac{\partial \hat{s}_t}{\partial \hat{\theta}_i^s} - A^t \hat{A}^{-t} \left[ \frac{\partial \hat{s}_t}{\partial \hat{\theta}_i^s} - \hat{M} \right], \tag{30}$$

where $M = \left( \sum_{k=0}^{t-1} A^k \right) C$ and $\hat{M} = \left( \sum_{k=0}^{t-1} \hat{A}^k \right) \hat{C}$. Note that we further have $\frac{\partial \hat{s}_t}{\partial \hat{\theta}_i^s} = \hat{M}$ according to Eq. (28). That is,

$$M \frac{\partial \theta_i^s}{\partial \hat{\theta}_i^s} = \frac{\partial s_t}{\partial \hat{s}_t} \hat{M}. \tag{31}$$

Recall that $\frac{\partial s_t}{\partial \hat{s}_t} = \mathbf{J}_t^h$ is full rank. Moreover, the full column rank of $M$ and $\hat{M}$ is guaranteed by the full rank of $A$ and the full column rank of $C$ (see Proposition 1). Therefore, we can derive

$$\text{rank}(M \frac{\partial \theta_i^s}{\partial \hat{\theta}_i^s}) = \text{rank}(\frac{\partial s_t}{\partial \hat{s}_t} \hat{M}) = \dim \theta_i^s. \tag{32}$$

Due to the rank inequality property of matrix products, we have

$$\text{rank}(M \frac{\partial \theta_i^s}{\partial \hat{\theta}_i^s}) \leq \min \left( M, \text{rank}(\frac{\partial \theta_i^s}{\partial \hat{\theta}_i^s}) \right) = \min \left( \dim \theta_i^s, \text{rank}(\frac{\partial \theta_i^s}{\partial \hat{\theta}_i^s}) \right) \leq \dim \theta_i^s. \tag{33}$$

Eq. (32) and Eq. (33) show that

$$\dim \theta_i^s \leq \min \left( \dim \theta_i^s, \text{rank}(\frac{\partial \theta_i^s}{\partial \hat{\theta}_i^s}) \right) \leq \dim \theta_i^s. \tag{34}$$

To make the above equation hold true, it must have $\text{rank}(\frac{\partial \theta_i^s}{\partial \hat{\theta}_i^s}) = \dim \theta_i^s$. That is, $\frac{\partial \theta_i^s}{\partial \hat{\theta}_i^s}$ must be full rank, then $\theta_i^s$ must be component-wise identifiable.

**Step 4: prove the identifiability of $\theta_i$ in the general case.**

This step can be easily demonstrated by directly combining the proofs above. $\square$

**Proposition 1.** *Suppose $A$ is a matrix with full rank, and $C$ is a matrix with full column rank. Define $M = \left( \sum_{k=0}^{t-1} A^k \right) C$. Then $M$ is full column rank.*

*Proof.* To establish that $M$ is full column rank, it suffices to show that $N = \sum_{k=0}^{t-1} A^k$ is of full rank. Assume the eigenvalues of $A$ are denoted by $\Lambda$. Given that $A$ is of full rank, the eigenvalues of $A^k$ are $\Lambda^k$. Let $\Lambda^N$ represent the eigenvalues of $N$, then we have

$$\Lambda^N = \sum_{k=0}^{t-1} \Lambda^k. \tag{35}$$

It is evident that there exists at least one $t$ such that for any non-zero $\Lambda$, Eq. (35) is non-zero, thereby confirming that $N$ has no zero eigenvalues. This verification ensures that $N$ is full rank. Hence, $M$ maintains full column rank. $\square$

### A.3 PROOF OF THEOREM 3

In Theorem 3, we foucus on the identifiablity of the newly added variables $s_t^{\text{add}}$ and their corresponding structural matrices, when the state space is expanded by incorporating additional dimensions.

**Theorem 3.** *(Identifiability of Expanded State Space). Assume the data generation process in Eq. (8). Consider the expansion of the state space $\mathcal{S}$ by incorporating additional dimensions. Suppose $s_t$ has already been identified according to Theorem 1, then the component-wise identifiability of the newly added variables $s_t^{\text{add}}$ and the additional structural matrices, i.e., $D^{s^{\text{add}} \rightarrow \cdot}$ and $D^{\cdot \rightarrow s^{\text{add}}}$, can be established if $s_t^{\text{add}}$ (1) represents a differentiable function of $[o_t, r_{t+1}]$, i.e., $s_t^{\text{add}} = f(o_t, r_{t+1})$, and (2) fulfills conditions (1) and (2) specified in Theorem 1.*

*Proof.* For the newly added variables, since they also satisfy the conditional independence condition, we can derive the same properties as described in Eq. (18) and Eq. (19) using the same technique in the proof steps of Theorem 1. Additionally, since the vector functions corresponding to $s_t^{\text{add}}$ also satisfies linear independence condition, it is straightforward that $s_t^{\text{add}}$ can also be identified component-wisely. As for the additional structural matrices introduced, the Markov condition and faithfulness assumptions required for their identifiability have already been demanded in the identifiability properties of the existing structural matrices, thus no additional proof is needed. □

**Corollary 1.** *(Identifiability under Multiple Shifts). Assume the data generation process in Eq. (20) involves both distribution shifts and state space shifts that comply with Theorem 2 and Theorem 3, respectively. In this case, both the domain-specific factor $\boldsymbol{\theta}_i$ and the newly added state variable $s_t^{add}$ are component-wise identifiable.*

*Proof.* This corollary can be directly derived by leveraging the conclusions from Theorems 1-3. □

### A.4    EXTENSION TO NONLINEAR CASES: CHALLENGES AND EMPIRICAL VALIDATION

The main challenge in extending the identifiability of $\theta_i^s$ to nonlinear scenarios lies in the fact that $g^s$, in this context, represents a general nonparametric transition dynamic. This makes it difficult to disentangle $\theta_i^s$ from $(s_{t-1}, \theta_i^s)$, as we do in the proofs of Theorem 2. Although recent works have made significant progress in establishing the identifiability of causal processes in nonparametric settings (Yao et al., 2021; 2022; Kong et al., 2023), they typically rely on the assumption of invertibility. However, as noted in Liu et al. (2023), while assuming the invertibility of the mixing function $g$ is reasonable, we cannot make the same assumption for $g^s$, as this often does not hold in practice. But this does not imply that $\theta_i^s$ is unidentifiable. On the contrary, the empirical results in Fig. 4 demonstrate that even in nonlinear settings, the learned $\hat{\theta}_i^s$ remains a monotonic function of the actual change factors $\theta_i^s$, corroborated by findings from Huang et al. (2021). This motivates us to extend Theorem 2 to broader nonlinear scenarios in our future research, a task that is challenging yet promising.

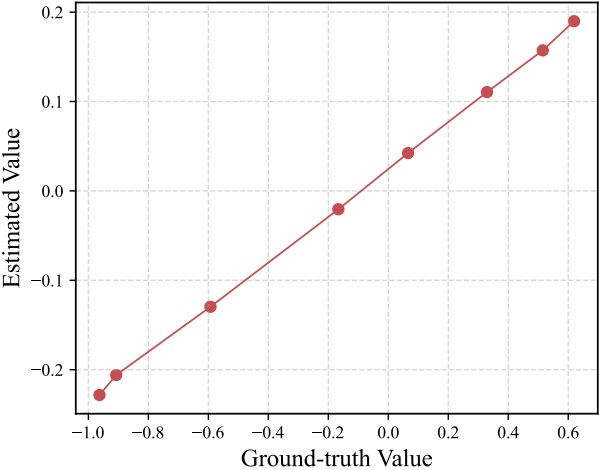

Figure 4: **The estimated $\hat{\theta}_i^s$ is a monotonic function of the ground-truth values in our simulations.**

## B    SELF-ADAPTIVE EXPANSION STRATEGY

We design three different approaches for state space expansion: *Random*, *Deterministic*, and *Self-Adaptive*. In *Random*, the number of causal variables expanded is randomly chosen. For *Deterministic*, we follow the approach used in DEN (Yoon et al., 2017), first adding a predefined number of variables to the causal graph and then applying group sparsity regularization to the network parameters corresponding to the newly added variables. Table 3 provides the final expansion results

| Experiments | *Random* | *Deterministic* | *Self-Adaptive* | Expansion Scope |
|---|---|---|---|---|
| Simulated | 4.0 | 5.0 | 4.2 | (0, 8] |
| CartPole | 6.2 | 6.0 | 4.0 | (0, 8] |
| CoinRun | 9.4 | 8.0 | 6.8 | (0, 10] |
| Atari | 7.8 | 8.0 | 6.4 | (0, 10] |

Table 3: Detailed expansion results of different methods in our experiments.

of various methods across these tasks, as well as the scope of expansion. Next, we introduce the *Self-Adaptive* method.

Different from prior methods, *Self-Adaptive* integrates state space expansion into the reinforcement learning framework. To achieve this, the first thing that needs to be done is to transform the expansion concept into a decision-making process. Since our goal is to determine how many causal variables should be incorporated into the causal graph, the action $u_t$ can be intuitively represented as the number of variables to add. Regarding the state variable $v_t$, it is designed to reflect the current state of the system. Given that the model's expansion is inseparable from its original structure and adaptability to the target task $\mathcal{M}_i$, we formulate the state variable as a reflection of both the original network size and its predictive capability on $\mathcal{M}_i$, denoted as $v_t = (x_t, \Delta_\tau)$. To be specific, $x_t = (x_t^o, x_t^r, x_t^s)$, where $x_t^o$, $x_t^r$, and $x_t^s$ represent the combination of the number of nodes for each layer in the models defined in Eq. (2), respectively. If the transition model is an $m$-layer network, then $x_t^s$ is an $m$-dimensional vector, with the $l$-th element representing the number of nodes in the $l$-th layer. Moreover, $\Delta_\tau = \tau - \tau^\star$ represents the difference between the model's predictive performance $\tau$ and the threshold $\tau^\star$.

Whenever the controller takes an action $u_t$, we correspondingly extend the model by augmenting it with additional components, and train the newly added parts from scratch with a few amount of data. For instance, if $u_t = d'$, it implies that $d'$ causal state variables will be incorporated. Then for the observation model, we only need to focus on learning the mapping from $s_t^{\text{add}}$ to $o_t$, together with the structural constraints $D^{s_t^{\text{add}} \to o}$. A similar principle applies to the reward and transition model. Finally, we re-estimate the performance of the expanded model, denoted as $\tau_t'$, and derive the reward as:

$$r_t = (\tau - \tau_t') - \lambda_r u_t, \tag{36}$$

where $\tau - \tau_t'$ reflects the change in the model's representational capacity before and after expansion, the term $-\lambda_r u_t$ acts as a regularization penalty that imposes a cost on model expansion, and $\lambda_r$ is the corresponding scaling factor, which is set to $0.01$ in our experiments.

Building upon the above foundation, it becomes feasible to develop and train a policy aimed at dynamically enhancing the causal model in adaptation to current task $\mathcal{M}_i$ through strategic expansion.

## C  DISTRIBUTION SHIFTS VS. SPACE EXPANSIONS IN A MDP SCENARIO

We further present a simple MDP scenario to illustrate the two types of environmental changes that CSR addresses: **distribution shifts** and **space expansions**. Specifically, Fig. 5(a) provides a graphical representation of the generative environment model for the source task, where $s_{1,t}$ denotes the latent causal variable, and $\theta = \{\theta^o, \theta^r\}$ represents task-specific change factors.

For **distribution shift** scenarios (Fig. 5(b)), the target task shares the same causal variable as the source task but differs in the value of $\theta$. For example, in CartPole, the gravity ($\theta^r$) might shift from 9.8 to 5. Notably, this implies that the causal diagram remains unchanged, an assumption commonly adopted in prior works (Huang et al., 2021; 2022; Gaya et al., 2022).

In contrast, **space expansion** involves the emergence of new variables (e.g., $s_{2,t}$ in Fig. 5(c)), which inevitably leads to changes in the causal diagram. Consequently, world models must expand their state and action spaces to accommodate these new variables. This necessity motivates the development of CSR. Algorithm 1 presents the pseudocode for CSR, where the model estimation and policy learning processes are implemented using the Dreamer framework. Notably, CSR is not restricted to Dreamer and can can also be implemented with a variety of policy-learning algorithms, such as Q-learning (Mnih et al., 2015) and DDPG (Lillicrap, 2015).

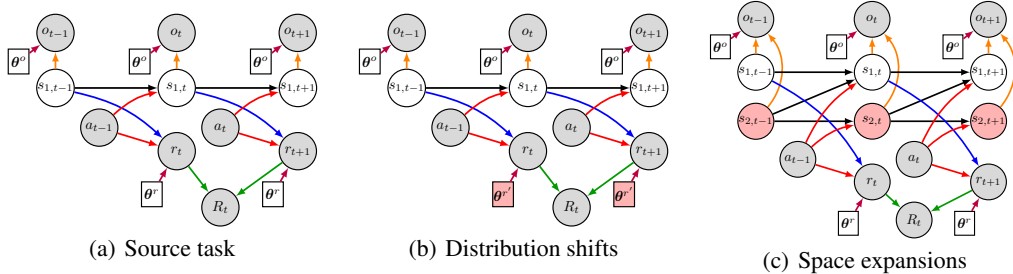

Figure 5: **A graphical illustration of the generative environment model and the two types of changes addressed by CSR.** (a) Source task; (b) Distribution shift scenario where the causal diagram remains unchanged but the value of $\boldsymbol{\theta}^r$ differs; (c) Space expansion scenario involving the emergence of new variable $\boldsymbol{s}_{2,t}$. Grey nodes denote observed variables, white nodes represent unobserved variables, and red nodes highlight the changing components in the target task compared to the source task.

---

**Algorithm 1:** Towards Generalizable RL through CSR

---

**Input:** Maximum distribution shifts detection step $T_c$.
Initialize World Model $W$ with parameters $\phi, \beta, \alpha$ randomly.
Initialize $D$ as an all-ones matrix.
Record multiple rollouts from source task $\mathcal{M}_1$ and estimate the model in Eq. (2).
Obtain the optimal policy $\pi^\star$ in $\mathcal{M}_1$ and calculate threshold $\tau^\star$ using $W$.
**for** *target tasks* $\mathcal{M}_i(i = 2, 3, \ldots)$ **do**
    Collect multiple rollouts $\mathcal{B}$ from $\mathcal{M}_i$.
    **while** *generalization* **do**
        // Model re-estimation
        **for** *training steps* $c = 1, \ldots, T_c$ **do**
            Draw data sequences $\{\langle o_t, a_t, r_t \rangle\}_{t \in \mathcal{T}_i}$ from $\mathcal{B}$.
            Compute model states $\boldsymbol{s}_t \sim q_\alpha(\boldsymbol{s}_t \mid \boldsymbol{s}_{t-1}, \boldsymbol{\theta}_i, a_{t-1}, o_t)$.
            Update $\boldsymbol{\theta}_i$ using $\mathcal{J}$, with all other parameters fixed.
        Calculate $\mathcal{L}_{\text{pred}}$ using $W$.
        // Distribution Shifts Detection and Characterization
        **if** $\mathcal{L}_{pred} < \tau^\star$ **then**
            **return** Latest model $W$ and policy $\pi^\star$ for task $\mathcal{M}_i$.
        **else**
            // State/Action Space Expansions
            Search to introduce new causal variables into the graph.
            **while** *not converged* **do**
                **for** *training steps* $c = 1, \ldots, C$ **do**
                    // Model estimation (Causal Graph Pruning is concurrently implemented)
                    Draw data sequences from $\mathcal{B}$ and compute model states using $q_\alpha$.
                    Update $\phi, \beta, \alpha, \boldsymbol{\theta}_i, D$ using $\mathcal{J}$.
                    // Policy Learning
                    Imagine trajectories from each $\boldsymbol{s}_t$.
                    Update policy $\pi^\star$ from the imagined trajectories via REINFORCE gradients.
                // Environment interaction
                **for** *time step* $t = 1, \ldots, T$ **do**
                    Select action $a_t$ with probability $\epsilon$; otherwise calculate $a_t$ using $\pi^\star$.
                    Execute action $a_t$ and receive reward $r_{t+1}$ and observation $o_{t+1}$.
                Store transition $\{\langle o_t, a_t, r_t \rangle\}_{t=1}^{T}$ into replay buffer $\mathcal{B}$.
            **return** Latest model $W$ and policy $\pi^\star$ for task $\mathcal{M}_i$.

# D    COMPLETE EXPERIMENTAL DETAILS

Below, we provide detailed implementation specifics for the experiments, including model architectures and training details, the selection of hyperparameters, a thorough description of the environments, and additional experimental results.

## D.1    MODEL ARCHITECTURES AND TRAINING DETAILS

**Model components.** Following Dreamer (Hafner et al., 2020; 2023), we implement the world model as a Recurrent State-Space Model (RSSM, (Cobbe et al., 2019)), the encoder and decoder in the representation model and observation model as convolutional neural networks (LeCun et al., 1989), and all other functions as multi-layer perceptrons with ELU activations (Clevert, 2015).

**The implementation of $D$.** We adopt the Gumbel-Softmax (Ng et al., 2022) and Sigmoid methods to approximate the binary masks $D$ in our experiments, which is a commonly used approach in causal representation learning.

**Training details.** During the generalization process, we use epsilon-greedy to balance the exploration-exploitation trade-off, and take straight-through gradients through the sampled representations for model estimation. Since the actions are always discrete, we adopt the REINFORCE gradients (Williams, 1992) with Adam optimizer (Kingma, 2014) for policy learning.

**Steps for distribution shifts detection.** Empirically, we set the maximum training steps for distribution shift detection as: 1k in Simulation, 2k in CartPole, 100k in Atari, and 250k in CoinRun.

**Training cost.** All experiments are conducted using an Nvidia A100 GPU. Training from scratch on the simulated and CartPole environments take less than 4 hours, training on Atari required approximately one day, and training on CoinRun takes about 4 days.

## D.2    HYPERPARAMETERS

| Simulated Environment | Architecture | Hyper Parameters |
|---|---|---|
| Change factor $\theta^{\boldsymbol{s}}$ | - | Uniform, [-1, 1] |
| Random noise $\epsilon_t^{\boldsymbol{s}}$ | - | Gaussian, $\mathcal{N}(0, 0.2I)$ |
| Reward function $g^r$ | Dense | 128, he uniform, relu |
|  | Dense | 64,   he uniform, relu |
|  | Dense | 1,    glorot uniform |
| Transition function $g^{\boldsymbol{s}}$ | Dense | 4,    glorot uniform, tanh |
| Observation function $g^o$ | Dense | 128, glorot uniform |

Table 4: Architecture and hyperparameters for the simulated environment.

| Hyper Parameters | Values in CartPole | Values in CoinRun and Atari |
|---|---|---|
| Action repeat | 1 | 4 |
| Batch size | 20 | 16 |
| Imagination horizon | 8 | 15 |
| Sequence length | 30 | 64 |
| Size of $\theta$ | 2 | 20 |
| Size of $h_t$ | 30 | 512 |
| Size of $z_t$ | 4 | 32 |
| Size of hidden nodes | 100 | 512 |
| Size of hidden layers | 2 | 2 |
| Regularization terms $\lambda_{\mathrm{KL}}, \lambda_{\mathrm{reg}}$ | 0.02 | 0.1 |

Table 5: Hyperparameters of CSR for CartPole, CoinRun and Atari games.

| Game | Modes | Difficulties |
|------|-------|--------------|
| Alien | [0, 1, 2, 3] | [0, 1, 2, 3] |
| Bank Heist | [0, 4, 8, 12, 16, 20, 24, 28] | [0, 1, 2, 3] |
| Crazy Climber | [0, 1, 2, 3] | [0, 1] |
| Gopher | [0, 2] | [0, 1] |
| Pong | [0, 1] | [0, 1] |

Table 6: Available modes and difficulties in each game of our Atari experiments.

### D.3 DETAILED DESCRIPTIONS OF THE ENVIRONMENTS

In this section, we provide detailed descriptions of the construction of these environments and present additional experimental results. For simulated experiments, we generate synthetic datasets that satisfy the two scenarios with different types of environmental changes. For CartPole, we consider distribution shifts in the task domains with different gravity or cart mass, and space variations by adding cart friction as the new state variable and additional force values for action expansion. For Atari games, we design the experiments by generating tasks with different game mode and difficulty levels. Such mode and difficulty switches lead to different consequences that changes the latent game dynamics or introduces new actions into the environment (Machado et al., 2018; Farebrother et al., 2018). For CoinRun, we train agents from easy levels and generalize them to difficulty levels where there could be new enemies that have never occurred before.

#### D.3.1 SIMULATED ENVIRONMENT

We construct the simulated environment based on the following POMDP framework:

$$
\begin{aligned}
s_1 &\sim \mathcal{N}(0, I_0), \\
o_t &= g^o(s_{t-1}), \\
s_t &= g^s(\theta^s, s_{t-1}, a_{t-1}) + \epsilon_t^s, \quad \epsilon_t^s \sim \mathcal{N}(0, I_\epsilon) \\
r_t &= g^r(s_{t-1}),
\end{aligned}
\tag{37}
$$

where $s_1$ and $\epsilon_t^s$ are sampled from Gaussian distributions, and functions $g^o$, $g^s$, and $g^r$ are implemented using MLPs. To simulate scenarios of distribution shifts, we generate random values for $\theta^s$ in different tasks. To model changes in the state space $\mathcal{S}$, we randomly augment it with $n$ dimensions, where $n$ is uniformly sampled from the range $[3, 7]$. Moreover, to introduce structural constraints into the data generation process, we initialize the network parameters for $g^o$, $g^s$, and $g^r$, by randomly dropping them out with a probability of $0.5$. The network weights then remain constant throughout the learning process. For each task, agents are allowed to collect data over 100 episodes, each consisting of 256 time steps. Table 4 provides the corresponding network architecture and hyperparameters.

#### D.3.2 CARTPOLE ENVIRONMENT

Based on the conclusions in Florian (2007), we modify the CartPole game to introduce changes in the distribution and state space. Specifically, for Task 1 and Task 2, the transition processes adhere to the following formulas:

$$
\begin{aligned}
\ddot{\psi} &= \frac{g \sin \psi + \cos \psi \left( \frac{-F - m_p l \dot{\psi}^2 \sin \psi}{m_c + m_p} \right) - \frac{\mu_p \dot{\psi}}{m_p l}}{l \left( \frac{4}{3} - \frac{m_p \cos^2 \psi}{m_c + m_p} \right)} \\
\ddot{x} &= \frac{F + m_p l \left( \dot{\psi}^2 \sin \psi - \ddot{\psi} \cos \psi \right)}{m_c + m_p},
\end{aligned}
\tag{38}
$$

where the parameters used are the same as those defined in Section 2 of Florian (2007), except that $\psi$ is used in place of $\theta$. By altering the values of $m_c$ and $g$, we can simulate distribution shifts. For Task 3, we introduce the friction between tha cart and the track into the game, denoted as $\mu_c$, thus

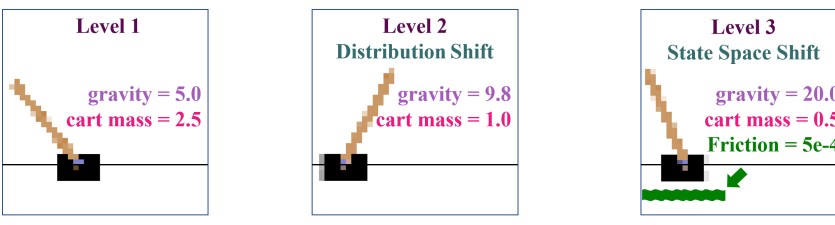

Figure 6: An illustration of the CartPole environment.

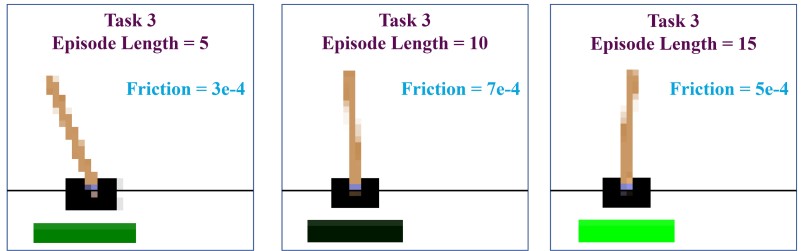

Figure 7: An illustration of the CartPole game under different friction coefficients.

altering Eq. (38) to Eq. (21) and Eq. (22) in Florian (2007), which is:

$$
\ddot{\psi} = \frac{g \sin \psi + \cos \psi \left\{ \frac{-F - m_p l \dot{\psi}^2 [\sin \psi + \mu_c \, \mathrm{sgn}(N_c \dot{x}) \cos \psi]}{m_c + m_p} + \mu_c g \, \mathrm{sgn}(N_c \dot{x}) \right\} - \frac{\mu_p \dot{\psi}}{m_p l}}{l \left\{ \frac{4}{3} - \frac{m_p \cos \psi}{m_c + m_p} [\cos \psi - \mu_c \, \mathrm{sgn}(N_c \dot{x})] \right\}}
$$

$$
\ddot{x} = \frac{F + m_p l \left( \dot{\psi}^2 \sin \psi - \ddot{\psi} \cos \psi \right) - \mu_c N_c \, \mathrm{sgn}(N_c \dot{x})}{m_c + m_p}.
$$

(39)

Note that $\mu_c$ varies cyclically every 5 steps among {3e-4, 5e-4, 7e-4}, so that the agent must continually monitor it throughout the process to achieve higher and stable rewards, This helps us assess whether the agent has detected the newly introduced variable. Additionally, we also visualize these changes in the image inputs; Fig. 7 presents examples under different friction coefficients. In the typical CartPole setup, the action values represent the direction of the force $F$. Specifically, 0 denotes a leftward force, while 1 indicates a rightward force, with a default magnitude of $F_{\mathrm{mag}} = 10$. In Task 4, we have expanded the possible values of $F$ to include $\{0.5 \times F_{\mathrm{mag}}, F_{\mathrm{mag}}, 1.5 \times F_{\mathrm{mag}}\}$, thereby extending the action dimension to 6. Our implementation is built upon Dreamer (Hafner et al., 2020), Table 5 lists the hyperparameters that are specifically set in our experiments. Fig. 8 illustrates the corresponding training results. Moreover, Fig. 9 shows a comparison of the reconstruction effects of different world models across the three tasks. Note that the transition model is an RSSM in our implementation. Consequently, we divide the state $s_t$ into a deterministic state $h_t$ and a stochastic state $z_t$. With this setup, the identified structural matrices in our experiment is shown in Fig. 10(a).

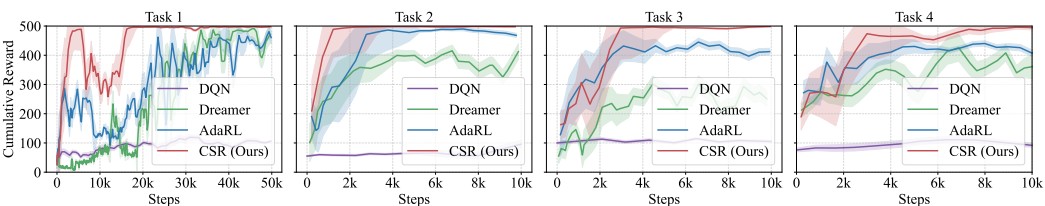

Figure 8: Training results of our CartPole experiments.

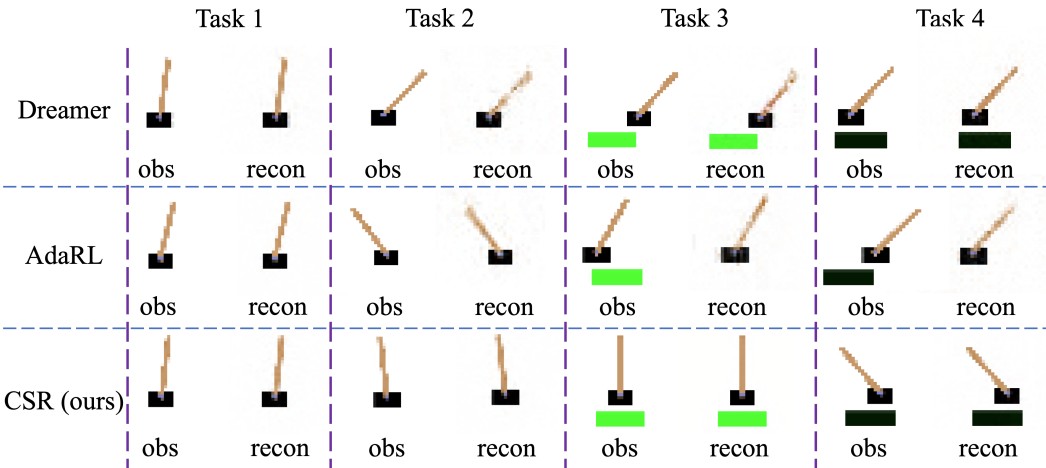

Figure 9: The reconstructed observations of different world models in CartPole.

### D.3.3 COINRUN ENVIRONMENT

CoinRun serves as an apt benchmark for studying generalization, owing to its simplicity and sufficient level diversity. Each level features a difficulty coefficient ranging from 1 to 3. Following Cobbe et al. (2019), we utilize a set of 500 levels as source tasks and generalize the agents to target tasks with higher difficulty levels outside these 500 levels. We maintain all environmental parameters consistent with those reported in Cobbe et al. (2019). For the world models of CoinRun and Atari games, we employ the same hyperparameters, which are listed in Table 5. Fig. 12 visualizes the reconstructions generated by various methods when generalizing to high-difficulty CoinRun games, where the first row displays the ground truth observations, the second row illustrates the model-generated reconstructions, and the third row highlights the differences between the ground truth and the reconstructions. We find that our proposed CSR method effectively captures newly emerged enemies, which the baseline methods fail to do. Moreover, Fig. 10(b) presents the estimated structural matrices.

### D.3.4 ATARI ENVIRONMENT

Atari serves as a classic benchmark in reinforcement learning, with most studies using it to evaluate the performance of proposed methods on fixed tasks. However, as mentioned in Machado et al. (2018), many Atari tasks are quite similar, also allowing for the assessment of a reinforcement learning method's generalization capabilities. Specifically, within the same game, we can adjust its modes and difficulty levels to alter the game dynamics. Although the goals of the game remain unchanged, increasing the complexity of modes and difficulty necessitates consideration of more variables, thus posing challenges for knowledge generalization.

According to Table 10 in Machado et al. (2018), we select five games that feature different modes and levels of difficulty, and set the task sequence as four in our experiments. The corresponding modes and difficulties available in these five games are given in Table 6. Fig. 11 gives an example in Crazy Climber. For Task 1, the agent is trained from scratch. For Tasks 2 to 4, different methods are employed to maximize the generalization of acquired knowledge to new tasks. Fig. 17 to Fig. 21 illustrates the training returns in these five games with different methods, Fig. 16 are the average generalization performances, and Fig. 15 are the corresponding reconstructions. Besides, we also illustrate how the structural matrices evolve during model adaptation in Fig. 10(c).

Note that changes in the latent state space in Atari games are not as straightforward as previous tasks, because variations in mode and difficulty typically influence latent state transitions rather than introducing new entities which are directly observable in the game environment. Hence, to further explore what the newly added variables represent, we first deactivate them and deduce their representations within the model, and then generate the reconstructions. Fig. 13 displays the reconstructed observations using the full state representations and after removing the newly added

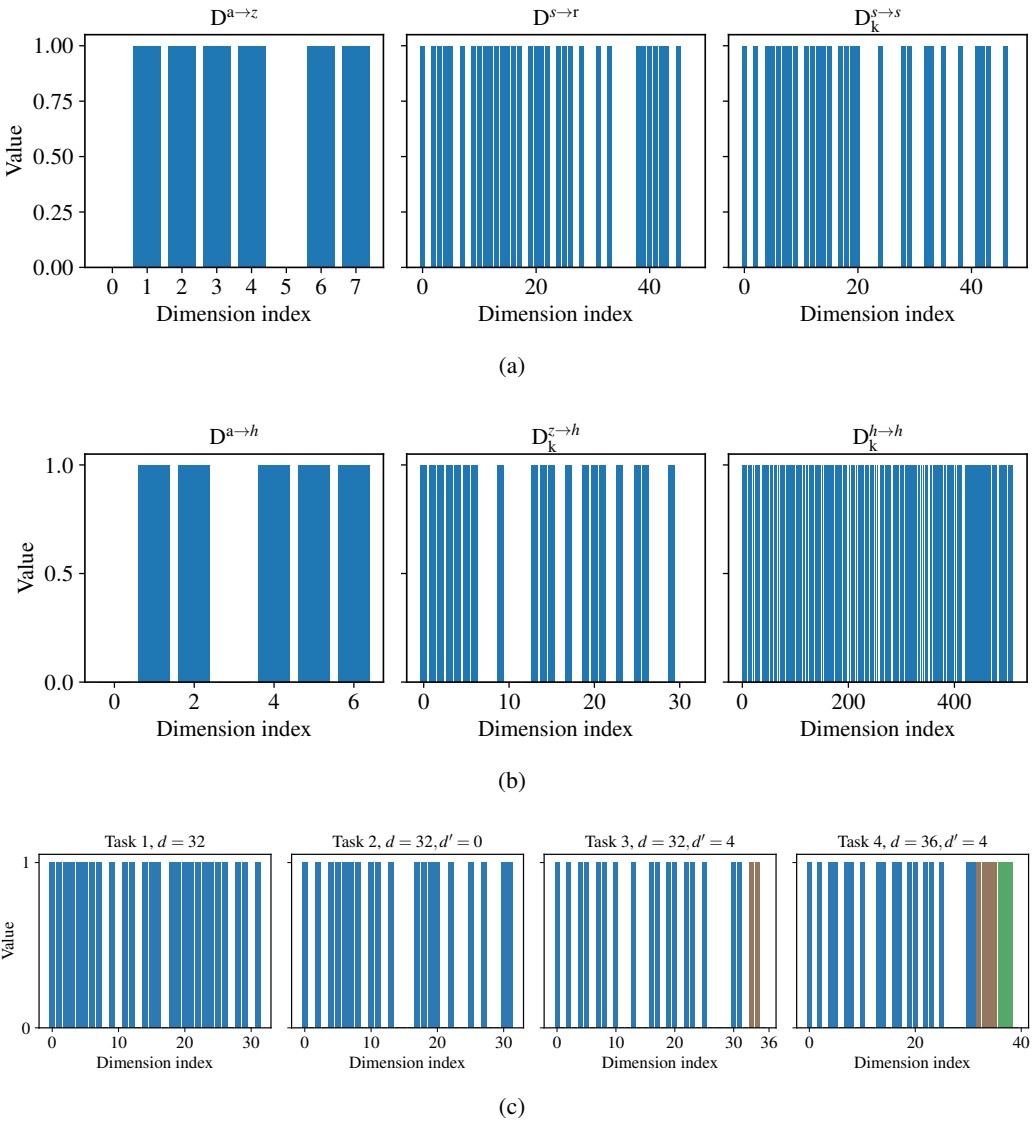

Figure 10: **Estimated causal structural matrices in the experiments:** (a) CartPole Task 1; (b) High difficulty CoinRun games; (c) An illustration of the state space in Pong and how $D_k^{z \to h}$ evolves across tasks. Here, $d$ represents the size of the state space, with $d' = 0$ indicating distribution shifts, and non-zero $d'$ signifying state space expansion.

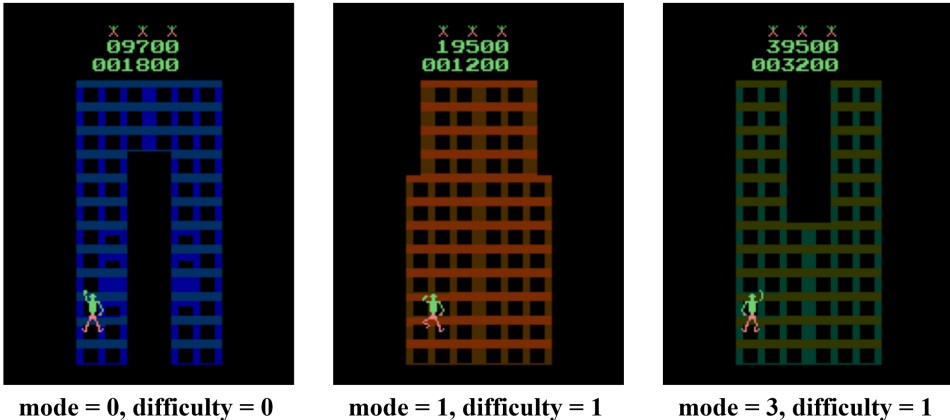

**mode = 0, difficulty = 0**    **mode = 1, difficulty = 1**    **mode = 3, difficulty = 1**

Figure 11: Different modes of the game Crazy Climber.

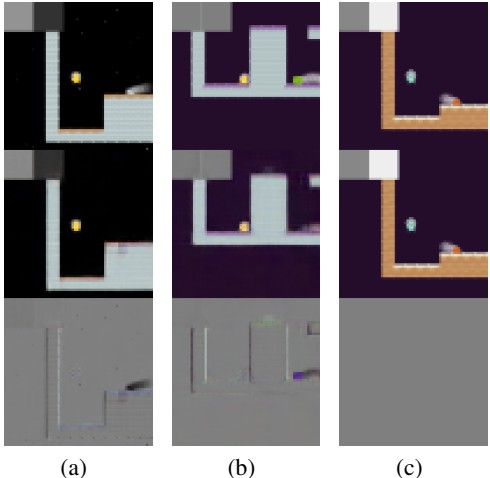

(a)          (b)          (c)

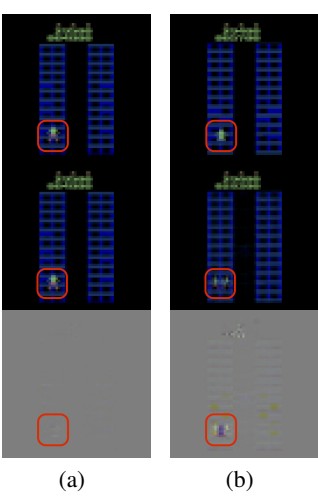

(a)          (b)

Figure 12: Visualization of reconstructions when generalizing to high-difficulty CoinRun games using various methods: (a) Dreamer; (b) AdaRL; (c) CSR (ours).

Figure 13: Visualized reconstructions in Atari target games using CSR with: (a) Full state representations; (b) Original state representations before expansion.

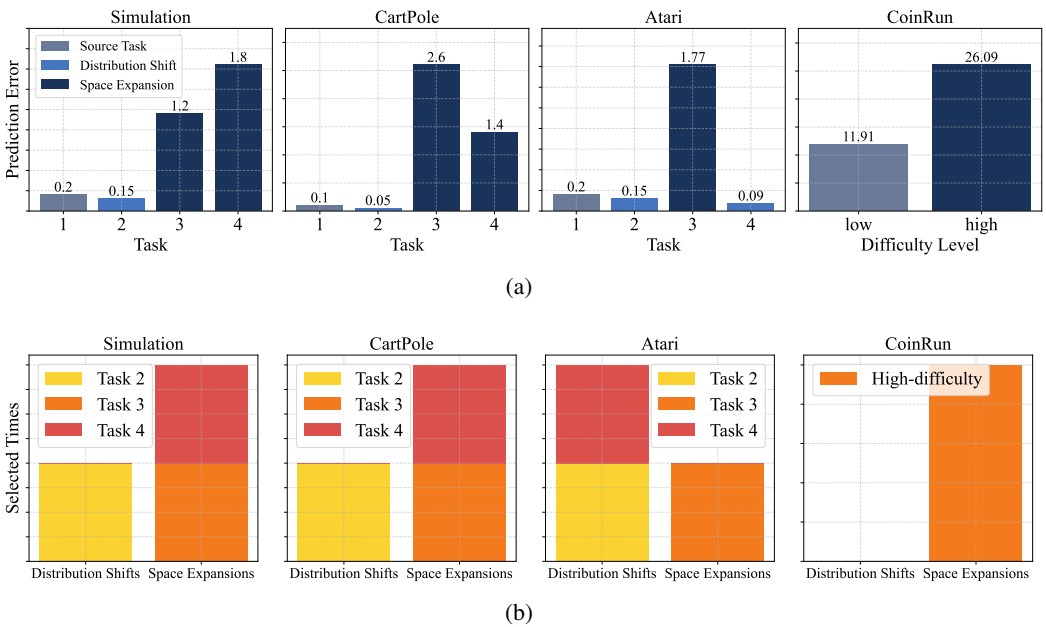

Figure 14: Visualization of (a) Evolution of prediction error $L_{\text{pred}}$ at the distribution shift detection step of CSR and (b) The ratio between the two modes of CSR across experimental environments.

variables. By comparing the observations before and after removal, it is evident that although the model can still reconstruct most of the buildings, it loses the precise information about the climber (the colorful person in the lower left corner of the image). Such disappearance in the reconstructions demonstrates the success of introducing the newly added variables in capturing the changing aspects in the latent state transitions. This further illustrates that CSR is also capable of handling general generalization tasks, even when the target domains do not exhibit significant space variations.

### D.4 ADDITIONAL EXPERIMENTAL RESULTS

To investigate the correlation between a task and the mechanism selected during adaptation, we further visualize the evolution of $L_{\text{pred}}$ and the ratio between the two modes of CSR (distribution shift detection and space expansion) across all experimental environments in Fig. 14. Note that for Atari games, averaging across environments does not yield meaningful insights. Therefore, we use Pong as a representative example, and set (mode, difficulty) sequentially across four tasks as (0,0), (0,1), (1,1), and (1,0). As seen in Task 2 of the simulated environment and CartPole, when only distribution shifts occur, $L_{\text{pred}}$ typically remains low. In contrast, the emergence of new variables, for example in Task 3 of Pong, consistently causes a substantial increase in prediction error—often by an order of magnitude or more. Besides, we also observe that the agent always opts to expand its causal world model to effectively address the added complexity and variability arising from the transition from simple to high-difficulty tasks in CoinRun. These observations collectively demonstrate that $L_{\text{pred}}$ serves as a reliable indicator of adaptation in our experiments.

## E A DISCUSSION ON FUTURE DIRECTIONS

While CSR presents a promising direction for extending RL to broader scenarios and has achieved meaningful progress, the pursuit of generalizable and interpretable RL still faces enduring challenges. In this section, we outline several potential research directions to inspire further innovative studies toward general artificial intelligence.

**Dynamic Graphs.** CSR focuses on domain generalization but has yet to address the challenges posed by nonstationary changes both over time and across tasks (Feng et al., 2022). It remains an

important direction for future research to develop methods that can automatically detect and model such changes to improve adaptability and robustness.

**Misalignment Problems.** The goal misalignment problem, also known as shortcut behavior, refers to a situation where an agent's performance appears aligned with the target goal but is actually driven by a side goal (Di Langosco et al., 2022; Delfosse et al., 2024). This often occurs because the target goal and the side goal share a common causal variable, defined as *Forks* in causal learning (Spirtes et al., 2001; Pearl & Mackenzie, 2018). Consequently, learning a causal world model defined in Eq. (2) that captures causal relationships rather than correlations could help mitigate this misalignment issue.

**Beyond Sequential Settings.** While our current focus is on task adaptation in sequential settings, CSR presents promising applications in continual reinforcement learning (CRL, (Khetarpal et al., 2022)), where the agent needs to utilize a replay buffer containing samples from both the current and previous tasks to identify the most similar task for policy transfer, or to address domain-agnostic settings by integrating domain shift detection techniques.

**Generalization across different games.** Another interesting direction for future research is the investigation of RL methods' ability to generalize across very different games, such as Space Invaders and Demon Attack. These games feature distinct visuals but share similar gameplay and rules. While humans can easily transfer knowledge between these tasks, this remains a challenging feat for artificial intelligence.

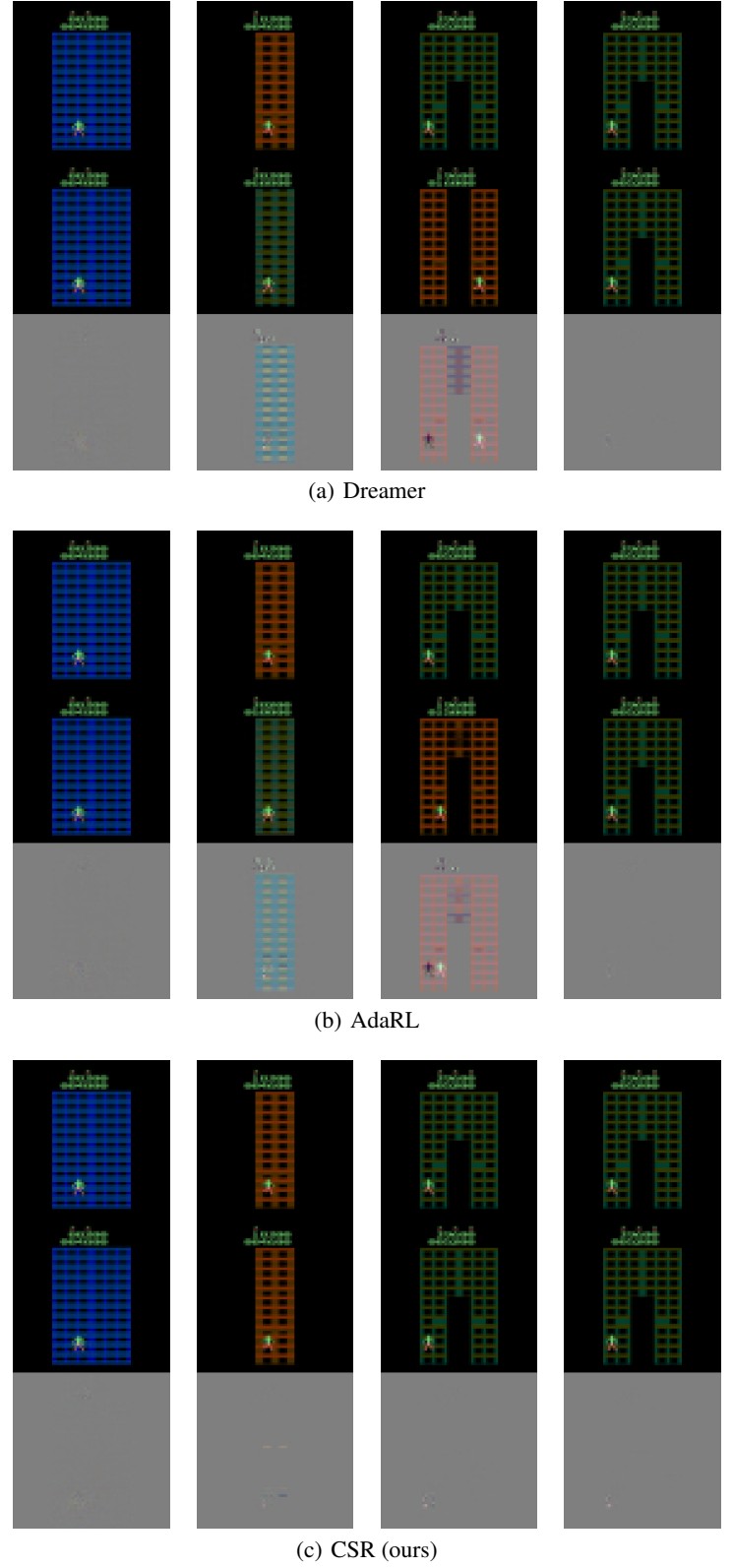

Figure 15: Visualization of reconstructions of various methods in the Atari games.

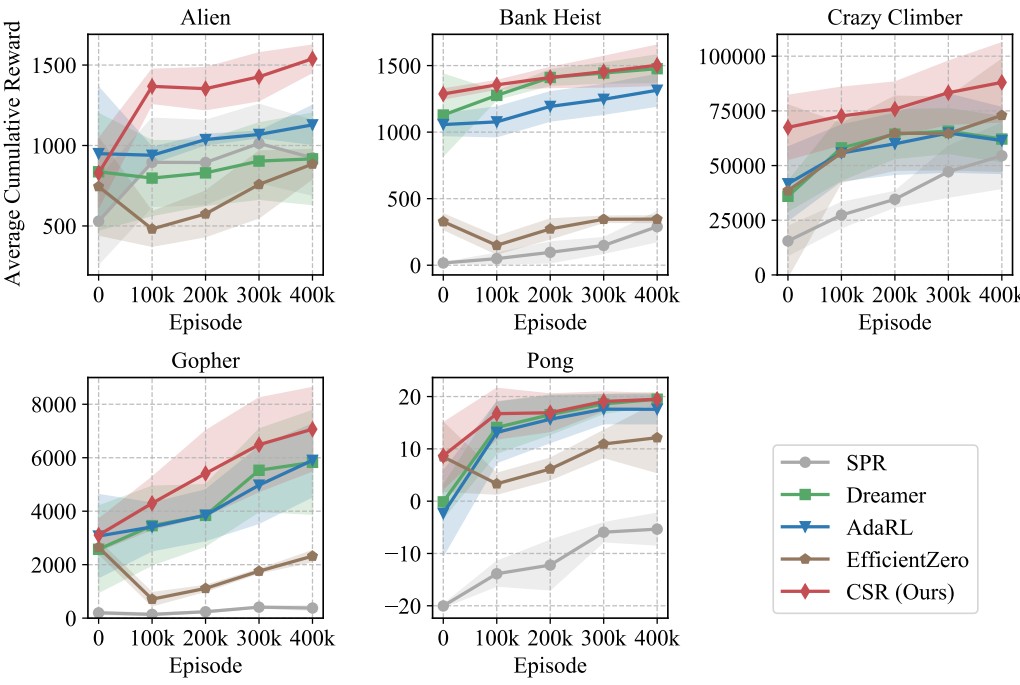

Figure 16: Average generalization performance of different methods in Atari games.

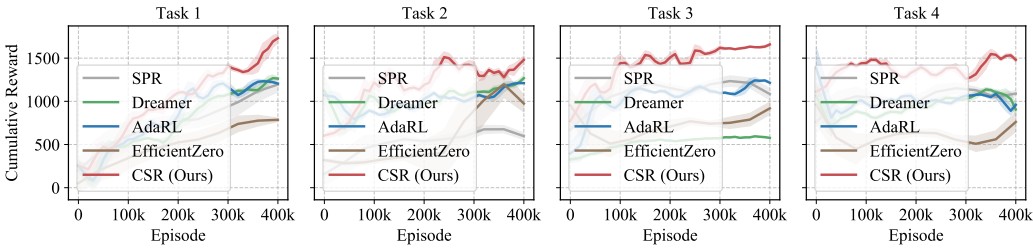

Figure 17: Training results of various methods in game Alien.

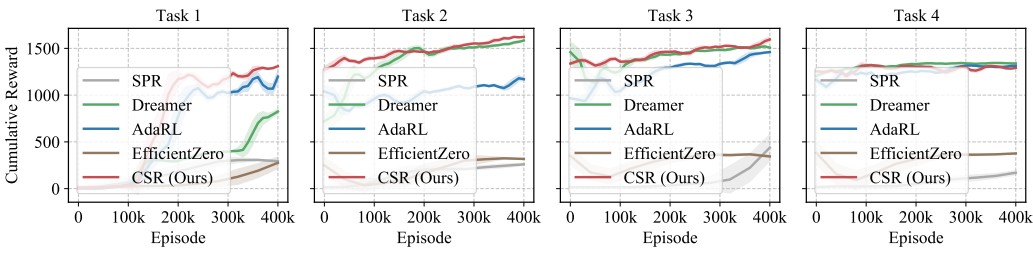

Figure 18: Training results of various methods in game Bank Heist.

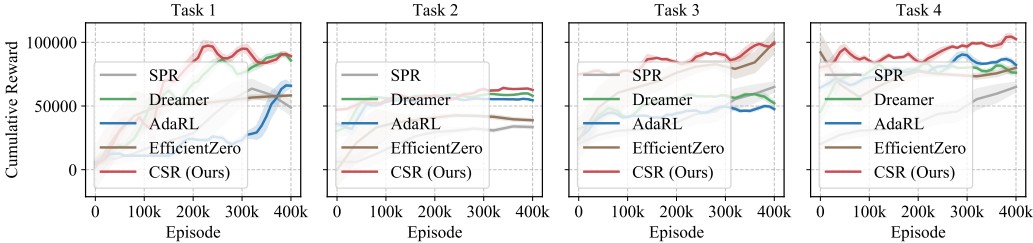

Figure 19: Training results of various methods in game Crazy Climber.

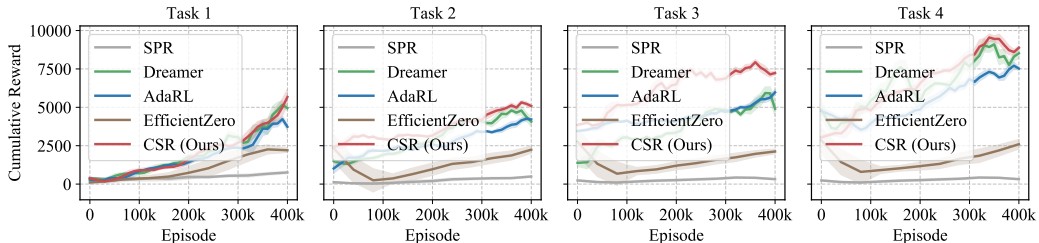

Figure 20: Training results of various methods in game Gopher.

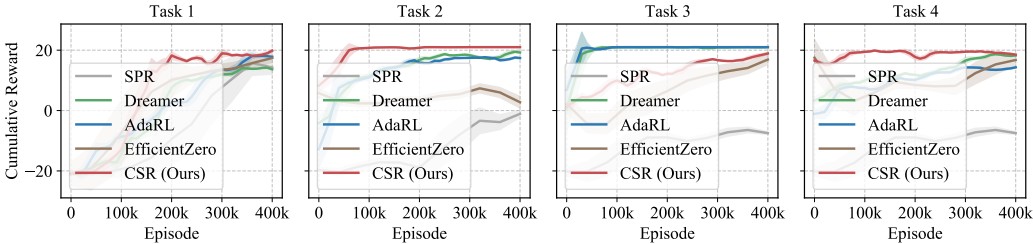

Figure 21: Training results of various methods in game Pong.

