# OpenReview forum: "Towards Generalizable Reinforcement Learning via Causality-Guided Self-Adaptive Representations"
_ICLR.cc/2025/Conference — ICLR 2025 Poster_

### Official Review · Reviewer_qQnp · 2024-10-27

**Soundness:** 3
**Presentation:** 3
**Contribution:** 3
**Rating:** 8
**Confidence:** 3

**Summary:**

The authors introduce a new algorithm, Causality-guided Self-adaptive Representation-based approach, or CSR. CSR aims to allow an RL agent to quickly adapt to novel tasks which are similar to previous tasks, but involve changes in various latent variables. These latent variables are divided into a state and a "task-specific change factor"; changes in these correspond to distributional shifts and shifts in the environmental structure, respectively.

CSR uses a world model, which consists of probabilistic mappings that approximate observations, rewards, transitions, and next-state inference functions. Within these models are various causal masks, which screen-off some conditioning variables, inducing a sparse causal structure.

Three theorems are given which show that the world models and their sub-components are identifiable under certain assumptions.

Details are then given for how CSR operates:
* How the world model is trained
* How distributional shifts are detected
* How the state-action space is modified when distributional shifts are insufficient to capture changes in observations

Lastly, experiments are done showing that, compared to various existing methods, CSR performs favorably following changes in the environment

**Strengths:**

Overall, I liked this paper a lot, and thought it contained many good ideas. I was impressed by the experimental results.

Originality:
- The core of the method proposed by the paper (uncovering structural relationships between latent variable to allow for quick adaptation to novel environments) is, to my knowledge, a new and creative contribution.
- Moreover, the algorithm is non-trivial, and involves multiple steps. The authors have solved numerous problems in order to create a pipeline which works.

Quality:
- The algorithm proposed by the paper, CSR, works well, and can outperform policy transfer with existing algorithms.
- The authors perform numerous experiments across different domains to demonstrate the performance of CSR.
- Solutions to various problems are not ad-hoc, and seem well-motivated.
- Finally, the authors provide multiple theoretical results which justify their approach.

Clarity:
- The writing was in general good, and the paper had a nice overall flow.

Significance:
- Transfer learning is an increasingly important topic in ML, and the authors do a good job of tackling the problem head-on.

**Weaknesses:**

The main weakness of the paper was that, at various points, the paper under-specified various implementation details for the algorithm. This significantly hurts the reproducibility of the work. Leaving various details under-specified in the main text would have been acceptable if the algorithm in the appendix (page 19, line 1013) filled in these details, and the main text included pointers to that appendix. Currently, the algorithm in the appendix is even less helpful for clarifying implementation details than the main text is! If the text and appendix are modified so that another research could *actually* reproduce the algorithm from the descriptions given, I think it likely I would increase my rating from borderline reject (5) to accept, good paper (8). A reader should not have to guess what you are doing - it ought to be clear from the text; otherwise it makes it very challenging for others to build upon your work. The method is very complicated - and while this is not a bad thing in and of itself, it does mean that care is needed to be clear about how the many aspects of the method work, and how they fit together. At present, that care has not been shown. If you are low on space in the main text then this is exactly the purpose of an appendix. My questions have illuminated at least some, but not all, of the points where implementation details are unclear for the paper.

(315) - "Let $d'$ denote the number of causal variables to be added, we first determine the value of $d'$ and introduce new causal features." This is not good grammar. Replace the comma with a period.

(319) - You say that the "existing parts of the world model are fine-tuned". Does your method suffer from catastrophic forgetting of previous tasks, or does it retain performance on task $M_1$ after going through tasks $M_2, \dots, M_9$ (for example)? Additionally, can you clarify why fine-tuning takes place, given that the previous model is intended to capture the existing relationships between variables?

(361) - You say that $D$ can "[transition] from $1$ to $0$ during model estimation". But earlier, in line (168), you say $D$ is a binary mask, i.e., each element is in $\{ 0, 1 \}$. It's unclear to me whether $D$ is a continuous or discrete variable, or whether you threshold $D$ at some point, and if so, at which point and how. Please clarify this, and (if necessary), remove the claim that $D$ is binary. Also, is $D$ constrained in $J_{\rm reg}$? Can it be negative?

(371) - The policy is conditioned on the state - how is the state generated? Presumably Thompson sampling through your world model. Is that correct? This detail should be in the paper.

(377) - "we train the newly added structure in the policy network while finetuning the original parameters". Train how? You haven't given sufficient details as to how you learn a policy. It looks deterministic, so is trained through DDPG? Or some other method? This is a very important detail!

A final note - it is not exactly clear to me where to position your method within the literature, or what problem it seeks to pose. At first, it seems as though it is a continual learning algorithm. But you don't test on a continual learning problem - you test on sequential learning. So is the imagined use-case for the algorithm simply adaptation to tasks in sequence?

Additionally, it appears that CSR requires notification whenever the task has changed. But this is surely a weakness compared to policy adaptation methods, which can just continue training on the incoming data without being informed of a distributional shift.

---------------------------------------------------------------------

As rebuttal to this review, the authors have made an effort to clarify the implementation details of the algorithm. I now feel comfortable saying that the paper could be reproduced by other researchers, and therefore view it as a solid contribution to the field.

**Questions:**

(140) - "In each period, only a replay buffer containing sequences from the current task is available, representing an online setting." I'm not sure I understand the reasoning here. If one maintains a replay buffer for each task, and tasks are encountered sequentially, why can't you access a replay buffer for a previous task during the current task?

(276) - In the expression for $J_{\rm rec}$, is there meant to be a log around the second probability term?

(276-283) - Can you clarify what exactly you mean by the expectation under $q_\alpha$ in the definition of $J_{\rm KL}$ and $J_{\rm rec}$? Importantly, are the $s_{t-1}$ and $s_t$ sampled afresh according to $q_\alpha$, or recalled from the memory replay buffer? Are gradients passed through these samples, or are they held fixed? Do you pass gradients through the samples using analytic expressions for the KL-divergence, via the reparametrisation trick, or by a REINFORCE/score-function style estimator?

(276-283) - I would also appreciate clarity on what the objectives are a function of. I'm assuming they're functions of $\phi, \beta, \alpha$, but they could also be functions of $\theta$. This just isn't clear from the main text.

(293) - "Therefore, in this step, we exclusively updates [sic.] the domain-specific part $\theta_i$, while keeping all other parameters unchanged". Can you be more specific about how these updates are done? Are they done using the objective $J$? How many updates are performed? Do you train to convergence? With early stopping? Using a train/validation split?

(303) - "Upon implementation, we use the final prediction loss of the model on the source task as the threshold value". This threshold seems to stringent in my mind. If there is some noise in $L_{\rm pred}$ (which would occur simply because the loss is sample based), then if $M_2 = M_1$, you would expect that on half of occasions the threshold would be violated, simply because of the noise. Am I missing something here? If not, can you confirm that this doesn't happen when $M_2 = M_1$.

---

> ### Author Response · Authors · 2024-11-20
>
> We sincerely appreciate your recognition of our work and efforts in providing valuable feedback, which significantly help improve its quality. Following your suggestions, we have enriched the appendix and Algorithm 1 with sufficient implementation details to facilitate reproduction by interested researchers. Furthermore, we commit to release the code upon publication, accompanied by detailed instructions for installation. Please see below for our responses to your questions.
>
> **(315).**
> We have corrected it as you suggested.
>
> **(319).**
> The original use of "finetune" was inaccurate. What we intended to convey is: "Accordingly, we focus on learning the newly incorporated components with only a few samples, as the previous model has already captured the existing relationships between variables." We have revised the sentence to clarify this.
>
> On the other hand, we believe that CSR can effectively address the issue of catastrophic forgetting by leveraging $\theta$. In other words, as long as the task-specific $\theta_{M_1}$ (e.g., $\theta_{M_1} = g = 9.8$ in Cartpole) has been learned and stored, even after encountering $M_2, \ldots, M_9$, where $\theta_{M_i}$ may take other values, policy adaptation can still be achieved by setting $\theta_M = 9.8$. This is because CSR updates only the components that have changed, without overwriting the original model.
>
> **(361).**
> In causal representation learning, $D$ represents binary masks that indicate the structural relationships among variables, and thus cannot take negative values. Upon implementation, we adopt the Gumbel-Softmax [1] and Sigmoid methods to approximate the binary masks and optimize them with $J$, which is a commonly used approach. Additionally, we do not threshold $D$, and the results demonstrate that the obtained values are effectively near 0 or 1.
>
> [1] Ng, Ignavier, et al. "Masked gradient-based causal structure learning." Proceedings of the 2022 SIAM International Conference on Data Mining (SDM). Society for Industrial and Applied Mathematics, 2022.
>
> **(371), (377) and (276-283).**
> As mentioned in line 271, CSR is built upon Dreamer, which iteratively performs exploration, model estimation, and policy learning (see Algorithm 1 of [2]). To maintain conciseness, some training details were not repeated in the main text, but we have now included them in Appendix D and the CSR algorithm. Below are our responses to your concerns regarding implementation details:
>
> - (371) We do not adopt Thompson sampling. Instead, for model estimation, latent state variables are inferred from pixel observations using an RSSM [3]. To balance the exploration-exploitation trade-off, we apply the $\epsilon$-greedy strategy. For policy learning, the trajectories are purely obtained through imagination that starts at the true model states $s_t$ of observation sequences and predicts both the actions and state values using the learned world model.
>
> - (377) The policy is learned using an actor-critic approach instead of DDPG. We use reinforce gradients to optimize the policy.
>
> - (276-283) The world model is trained by randomly sampling from the memory replay buffer that contains sequences of observation $o_t$,  actions $a_t$ and rewards $r_t$. We use the representation model $q_\alpha$ to encode categorical latent state $s_t$ from the sampled sequences, and take straight-through gradients through the sampling step (see DreamerV2 [4]).
>
> [2] Hafner, Danijar, et al. "Dream to control: Learning behaviors by latent imagination." arxiv preprint arxiv:1912.01603 (2019).
>
> [3] Hafner, Danijar, et al. "Learning latent dynamics for planning from pixels." International conference on machine learning. PMLR, 2019.
>
> [4] Hafner, Danijar, et al. "Mastering Atari with discrete world models." *arXiv preprint arXiv:2010.02193*, 2020.

---

> ### Author Response · Authors · 2024-11-20
>
> **Position within the literature.**
> Yes, our current focus is on task adaptation in sequential settings. Nonetheless, CSR holds significant potential for applications in the continual learning (CL) domain. Actually, we have previously investigated several CRL benchmarks, such as Brax [5], CausalWorld [6], Continual World [7], and CORA [8]. However, these benchmarks generally assume a consistent environment space across tasks [9], which contrasts with the setting of SCR. Consequently, we have not conducted tests within the continual learning context. We plan to extend CSR to CRL scenarios and adapt these benchmarks for testing as part of our future work. Thank you very much for pointing this out!
>
> [5] Freeman, C. Daniel, et al. "Brax--a differentiable physics engine for large scale rigid body simulation." arXiv preprint arXiv:2106.13281 (2021).
>
> [6] Ahmed, Ossama, et al. "Causalworld: A robotic manipulation benchmark for causal structure and transfer learning." arXiv preprint arXiv:2010.04296 (2020).
>
> [7] Wołczyk, Maciej, et al. "Continual world: A robotic benchmark for continual reinforcement learning." Advances in Neural Information Processing Systems 34 (2021): 28496-28510.
>
> [8] Powers, Sam, et al. "Cora: Benchmarks, baselines, and metrics as a platform for continual reinforcement learning agents." Conference on Lifelong Learning Agents. PMLR, 2022.
>
> [9] Khetarpal, Khimya, et al. "Towards continual reinforcement learning: A review and perspectives." Journal of Artificial Intelligence Research 75 (2022): 1401-1476.
>
> **Adapting CSR to domain-agnostic scenarios.**
> Yes, CSR is currently designed for a domain-aware setting. However, it can easily handle domain-agnostic settings by incorporating domain shift detection techniques, as demonstrated in [10]. During the method design process, we also explored and experimented in this direction but ultimately chose not to include this component to avoid overcomplicating the CSR pipeline with excessive sub-components. A potential future research direction could be the seamless integration of distribution shift detection with domain shift detection. Thank you once again for your thoughtful and valuable insights!
>
> [10] Wang, Zhenyi, et al. "Learning to learn and remember super long multi-domain task sequence." Proceedings of the IEEE/CVF Conference on Computer Vision and Pattern Recognition. 2022.
>
> **Expanding CSR beyond sequential settings.**
> CSR can be naturally extended to settings where the agent leverages a replay buffer not only from the current task but also from previous tasks. However, as it represents an initial exploration of generalizable RL by learning causal world models, we currently focus on a simplified sequential setting where the agent concentrates on domain adaptation in a single target task. We look forward to further enhancing CSR in future work to unlock its potential in more complex scenarios like CRL.
>
> **(276).**
> Yes, we have corrected this typo.
>
> **(276-283).**
> $J$ are functions of $\psi$, $\beta$, $\alpha$, $\theta$, and $D$ because these variables are jointly optimized through $J$.
>
> **(293).**
> Yes, we utilize $J$ to update $\theta$. To be specific, we empirically set the maximum training steps for distribution shift detection as: 1k in Simulation, 2k in CartPole, 100k in Atari, and 250k in CoinRun.
>
> **(303).**
> Empirically, we observe that such incorrect detections do not occur in our experiments. This is because introducing a new state typically results in a substantial increase in prediction error (at least twice as much), even with minimal changes in dimensionality. Furthermore, we have also illustrated the evolution of $L_\text{pred}$ across tasks in Fig. 14, which further supports this observation.

---

> ### Author Response · Authors · 2024-11-25
>
> Dear Reviewer **qQnp**,
>
> Thank you for your thoughtful feedback and for highlighting the importance of detailed implementation in our paper, which has greatly helped us improve our work. Following your suggestions, we have included additional implementation details in Appendix D and also in Algorithm 1 to facilitate reproducibility and to inspire further advancements in this area.
>
> As the discussion phase is coming to an end, we kindly ask whether our response has resolved your main concerns. If there are any remaining issues, please let us know, and we are willing to address them.
>
> If your concerns have been addressed, we would deeply appreciate it if you could consider updating your recommendation accordingly. We value your time and insights and look forward to your response.
>
> Thank you again for your attention to our work.
>
> Sincerely,
>
> Submission620 Authors

---

> > ### Comment · Reviewer_qQnp · 2024-11-26
> > **Great work!**
> >
> > Having reviewed the changes made by the authors to increase the clarity of the work, I now feel comfortable recommending that this paper be accepted to ICLR. I would like to congratulate the authors on their paper; I hope that it is accepted to the conference.

---

> > > ### Author Response · Authors · 2024-11-26
> > > **Thank you for your response!**
> > >
> > > Thank you once again for your time devoted to our paper and your invaluable feedback! Your valuable comments have helped improve our presentation a lot. Thank you very much!

---

### Official Review · Reviewer_jHns · 2024-11-02

**Soundness:** 4
**Presentation:** 3
**Contribution:** 3
**Rating:** 8
**Confidence:** 4

**Summary:**

The authors introduce *causality-guided self-adaptive representation* (CSR), for RL agents. It introduces a causal world model into RL agents, that allow them to distinguish two types of environmental changes:
1. distribution shifts
2. State/Action space expansion.
The causal world model generates the next observation and can be adjusted if the predicted observation differs too much from the observed one, allowing to detect that the environment has changed.

Thus, the authors evaluate their baselines on different sets of environments such as the famous Atari and Procgen.

**Strengths:**

**Interesting and novel motivation**. The idea to use causality to distinguish different types of changes is interesting, and might be one of the most promising directions to allow RL agents to generalize past their training environments.

**Soundness of the method**. The method is overall sound.

**Overall clarity**. The overall structure of the paper makes it quite easy to follow and understand the method. The clarity ~can be~ has been improved (as detail led below).

**Broad experimental evaluation**. The experimental evaluation is broad (Procgen+Atari), and demonstrate the superiority of CSR in terms of pure performances to the baselines. The Atari modifications of the environments have been introduced in [1], it would be nice to cite them.

[1] Farebrother, J., Machado, M. C., & Bowling, M. Generalization and regularization in DQN. arXiv (2018).

**Weaknesses:**

Several points are unclear:
* **Implicit world models** CSR still encodes an implicit world model. The learned world models cannot be interpreted. The use of causality had me first think of the contrary. It could be nice to insist on this point. Explicit RL reasoning agents have also been developed ([1, 2, 5]).
* **The difference between the two types of change is not clear**. Beside the apparent effort of the authors to explain this point, I'm still unsure of the difference, and it's a core contribution. I would advise using the MDP in the introduction, explaining how the MDP is modified for this type of change.
* **What are the answered scientific questions?** The experimental evaluations can be reorganized in terms of scientific questions (Q1 learning ability, Q2 adaptivity, Q3 core components for performances, i.e. ablation)...

A potentially striking point for the paper would be:
Could a causal world model help mitigate misgeneralisation/misalignment problems ([2, 3]) ? The misalignment detected in [2] on the *LazyEnemy* variation of Pong [4] (where they show that RL agents tend to follow the vertical position of the enemy instead of the ball) is quite striking. A causal world model, that could then update itself to detect that the Enemy is following the ball instead of the contrary could there help, couldn't it ?
A discussion on this issue or its use in introduction/future work could demonstrate the potential benefit of causal WM in RL.

------------
* [1] Luo et al. "INSIGHT: End-to-End Neuro-Symbolic Visual Reinforcement Learning with Language Explanations." ICML (2024).
* [2] Delfosse et al. "Interpretable concept bottlenecks to align reinforcement learning agents." Advances in Neural Information Processing Systems 37 (2024).
* [3] Di Langosco et al. "Goal misgeneralization in deep reinforcement learning." International Conference on Machine Learning. PMLR (2022).
* [4] Delfosse et al. "HackAtari: Atari Learning Environments for Robust and Continual Reinforcement Learning." arXiv (2024).
* [5] Marton, et al. "SYMPOL: Symbolic Tree-Based On-Policy Reinforcement Learning." arXiv (2024).

I am willing to augment my scores if these are addressed.

**Questions:**

* how does CSR compare with the other methods in terms of added parameters and training/inference time ?

---

> ### Author Response · Authors · 2024-11-20
>
> We appreciate your thoughtful comments and time devoted to reviewing this paper. We have added the citation as suggested and hope the following response properly addresses your concerns.
>
> **Implicit world models.**
> Thank you for raising this point. But we would like to clarify that CSR indeed encodes a world model that is designed to be interpretable. By discovering and utilizing causal graphs and their corresponding causal matrices $D$, CSR can effectively identify the underlying causal relationships in each task, thereby assisting in interpretable decision-making, as in [1, 2, 5]. Besides, in contrast to neuro-symbolic approaches, one of CSR’s key advantages is that it does not require predefined symbols, enabling it to be applied more generally across various domains.
>
> For example, consider Fig. 5(a), where the causal graph represents a simple MDP scenario. In Cartpole, $s_1$ could represent the cart position, and $\theta^o$ denotes the background color. Based on the causal graph, we observe that $D^{s_1 \to r} = 1$ and $D^{\theta^o \to r} = 0$, revealing that "policy learning needs to consider the cart position as it influences the reward, but does not need to account for the background color, as it is an irrelevant variable." The agent uses this rule to guide its decisions, which is precisely the approach employed in Sections 3.3 and 3.4.
>
> Moreover, CSR can also explain its adaptation mechanism (i.e., why a distribution shift detection or space expansion is chosen). Referring to Fig. 5(b), after re-estimating the values of $\theta^{r'} = \{g\}$ in Task 2, CSR clarifies that the reason for not selecting space expansion is: "Only the gravity ($g$) have changed in the environment, with no new variables introduced, thus no expansion is necessary."
>
> We further illustrate in Fig. 10 the estimated causal matrices from our experiments, which reflect which variables are significant and explain why they are chosen for policy learning. It is important to note that unlike the MDP example above, all of our experiments involve POMDP scenarios, where latent state variables are learned from pixel images. Thus, directly determining the representation of each variable is not possible. However, a more intuitive approach is to observe the activation or deactivation of causal variables and analyze the corresponding changes in the reconstructed image, specifically noting which parts of the image appear or disappear. Fig. 13 provides an example from the Crazy Climber game, showing the meaning of expanded variables and explaining why the expansion is necessary. A detailed analysis can be found in Appendix D.3.4.
>
> **The difference between the two types of change is not clear.**
> Thank you for your feedback. We have added a simple MDP example with a graphical illustration in Appendix C to clarify the difference between the two types of change.
>
> **What are the answered scientific questions?**
> Thank you for the valuable suggestion. We have reorganized the experimental evaluations around the following scientific questions:
>
> - **Q1:** Can CSR effectively detect and adapt to the two types of environmental changes?
> - **Q2:** Does the incorporation of causal knowledge enhance the generalization performance?
> - **Q3:** Is searching for the optimal expansion structure necessary?
>
> **Could a causal world model help mitigate misgeneralisation/misalignment problems?**
> Yes! CSR can indeed help mitigate misgeneralization problems. The misalignment you mention can be traced back to the distinction between correlation and causality. For instance, in the LazyEnemy variation of Pong, the causal graph is: Y (enemy position) $\leftarrow$ X (ball position) $\rightarrow$ Z (agent position). It is clear that there is no causal edge between Y and Z, as the agent position is not directly caused by the enemy’s position, but by the ball position, which aligns with the goal. In this structure, X is a common cause of both Y and Z, creating a correlation that leads to the goal misalignment problem [6, 7]. CSR can address this by learning a causal world model that identifies causal edges rather than correlations. We have discussed this further in Appendix E. Thank you very much for this insightful point!
>
> [6] Pearl, J. , Mackenzie, D. . (2018). The book of why : the new science of cause and effect. Science, 361(6405), 855.2-855.
>
> [7] Spirtes, P.; Glymour, C.; and Scheines, R. 2001. Causation, Prediction, and Search. Cambridge, MA: MIT Press, 2nd edition.

---

> > ### Author Response · Authors · 2024-11-20
> >
> > **how does CSR compare with the other methods?**
> > To ensure a fair comparison, we have kept the parameter sizes consistent across all methods (e.g., other baselines match CSR’s state size after expansion). CSR’s training time is consistent with Dreamer’s (lines 1157-1159), excluding the additional time for searching the optimal expansion structure, which depends on the chosen expansion strategies. As mentioned in lines 335–337 and 492–498, CSR allows flexibility in selecting expansion strategies, with a trade-off to consider in practical applications.

---

> > > ### Comment · Reviewer_jHns · 2024-11-22
> > > **Nice rebuttal, one more thing...**
> > >
> > > I want to thank the authors for considering the feedback provided by every reviewer. I think their rebuttal deserves to be considered.
> > >
> > > First, I want to highlight that I really appreciate the answers given to the other reviewers, and paticularly the additional experiments conducted in the Atari domain.
> > >
> > > I strongly disagree with reviewer **Pjxf** that states that a weakness of the paper is the non applicability of causal models to complex or noisy environments. Sure, causal discovery can be error-prone, so were RL methods in the early days, and they can still be.
> > >
> > > **This work is novel, well motivated, provides strong empirical evidence, and does not need to solve AGI to be accepted at ICLR.**
> > >
> > > I find the new structure of the experimental section way easier to quickly grasp.
> > >
> > > Let me provide several more small concerns I have:
> > > * The captions of the figures and table can be improved to again improve clarity. I think that if you structure the captions with:
> > >   * The first sentence that provide the general message (in bold).
> > >   * The second one explaining what is depicted.
> > >   * The following ones that provide additional details and references to *e.g.* appendices.
> > >
> > > For example, the caption of Figure 1 could be: **Environmental change may or may not necessitate RL agents retraining**, as illustrated on different variations of the Coinrun game. The changes from (a) to (b), in the amount and shape of obstacles do not prevent the agent from completing the task, while the changes introduced in (c) (*i.e.* deadly holes and enemies) require retraining. This should make the difference between the two types of changes understandable by readers that take a quick look at your paper, before deciding if they actually should read it (and I think they should read it!).
> > > For Table 1: **CSR is the only method that can consistently adapt to successive environmental changes in CartPole.** Evaluation results of different approaches, with a maximum episode length of 500. ...
> > > For Table 2: **Our CSR method outperform every evaluated model-based method on the selected set of Atari games. ...**
> > > ...
> > >
> > > * If I understood the example of the appendix, CSR still provide an implicit world model by design. You are then applying an interpretation to this world model. Knowing the experimenter bias, CSR could have learned a wrong causal world model (*e.g.* relying on the wrong variables), and you might interpret it to be right). Again, your method does not have to address all raised points (and you don't have to claim it). The non-reliance of symbols of your method completely justifies its implicit design choice.
> > > * Related to that and because you already went through the trouble of training agents on the *Pong*.
> > > 1) Could you provide the learned causal WM on this environment ?
> > > 2) Would it be complicated to test your agent in the *LazyEnemy* version (of HackAtari) ? If your agent is misaligned and can actually detect through the change in the environment that the enemy could avoid following the ball, it would be a very striking example of the power of CWM to solve misalignments.
> > >
> > > I am already increasing my score because of the clarity improvement and of these additional experiments.

---

> ### Author Response · Authors · 2024-11-23
> **Thank you for reviewing our rebuttal !**
>
> Thank you for your thorough review of our rebuttal and for your appreciation of our efforts! Below are the responses to the remaining concerns you have raised.
>
> **The captions of the figures and table can be improved to again improve clarity.**
> Thank you for your valuable suggestions. We have revised the captions as you recommended and highlighted the changes in orange for easier identification.
>
> **An implicit world model.**
> Yes, your understanding is correct. The CSR agent cannot directly explain the reason for each action choice because it is not feasible to determine precisely what each latent state, inferred from pixel images, represents. In light of your comment, moving forward, we will explore how to enable the model to autonomously identify the meanings of each state, thereby revealing the reasoning behind each decision. Another potential benefit of this approach is that it allows us to determine the minimal sufficient state representation for each task, rather than empirically choosing a suitable value for the state dimension and then pruning them as we currently do, which could help reduce the network size of the model. This is indeed an invaluable point; we thank you again for highlighting these considerations.
>
> **The learned causal WM on Pong.**
> Yes. We have illustrated the state space in Pong and the evolution of the graphical structure $D_k^{z \to h}$ across tasks in Fig. 10(c), where $z$ represents the latent state variable and $h$ is the deterministic state variable in our implementation based on a RSSM [8] world model.
>
> [8] Hafner, Danijar, et al. "Learning latent dynamics for planning from pixels." International conference on machine learning. PMLR, 2019.
>
> **Test CSR in the LazyEnemy version.**
> It may not be as straightforward as it seems. After reviewing the HackAtari code, we identified a number of differences between HackAtari and our implementation in handling the Gym Atari environment, particularly in the *step()*, *reset()*, and certain *gym.Wrapper* classes. These differences make it challenging to directly test CSR in the LazyEnemy version. We are actively exploring solutions and will do our best to complete the testing within the discussion period, though we cannot fully guarantee this timeline. Updates will be shared as soon as the results are available. Thank you for your understanding and for this valuable suggestion.
>
> Please let us know if you have any further concerns. Thank you once again for your time and consideration!

---

> > ### Author Response · Authors · 2024-11-26
> > **Update testing results in the LazyEnemy version.**
> >
> > We have successfully tested CSR and Dreamer when trained on the original Pong and tested on LazyEnemy. The results are as follows:
> >
> > |                                |  CSR     |  Dreamer | PPO (reported in [4]) |
> > |---------------------------------|:--------:|:--------:|:---------------------:|
> > | zero-shot                      | **-5.53** |  -15.03  |         -12.6         |
> > | few-shot (100k)                | **16.51** |  12.91   |           \            |
> >
> >
> > - **CSR can help mitigate misalignment problems**: We are surprised to find that world models seem more prone to misalignment problems compared to PPO, particularly in the zero-shot setting. Notably, CSR showed a 63.2% improvement in generalization performance over Dreamer in the zero-shot case, and a 27.8% improvement in the few-shot case. This suggests that learning a causal world model can help alleviate misalignment issues to some extent.
> >
> > - **Zero-shot performance still has room for improvement**: In misalignment problems, the environment and game goals do not change; the issue stems from deficiencies in the agent itself. Therefore, zero-shot performance is of greater importance and still holds promise. Unfortunately, despite CSR’s strong generalization ability in the few-shot scenario, its zero-shot performance remains suboptimal. We attribute this to the implicit world model it characterizes, but remain confident in CSR's potential when it is capable of explaining each decision it makes.

---

### Official Review · Reviewer_PJxf · 2024-11-03

**Soundness:** 2
**Presentation:** 3
**Contribution:** 1
**Rating:** 6
**Confidence:** 5

**Summary:**

This paper proposes a novel model, CSR to enhance RL generalization across diverse and dynamic environments. CSR tackles distributional changes and environmental space variations, such as introducing new elements or challenges. CSR uses causal representation learning to identify latent variables and structural relationships, enabling agents to recognize and respond to environment changes more efficiently. The approach is structured around a three-step process: detecting shifts, expanding the model when necessary, and pruning irrelevant variables for task-specific learning. Experimental results show that CSR achieves faster adaptation and better performance than existing methods on tasks like CartPole, CoinRun, and Atari games.

**Strengths:**

- This paper is well-written. The structure of CSR is clearly explained, and the authors provide some technical details.

- The three-step CSR framework (detection, expansion, and pruning) is well-structured. It provides a systematic way to handle different environment changes, contributing to effective and efficient policy adaptation.

- Including ablation studies and comparative evaluations for various expansion strategies (random, deterministic, self-adaptive) provides valuable insights into the effectiveness of different design choices.

**Weaknesses:**

- CSR's effectiveness depends on accurately identifying latent causal variables and relationships. In noisy or complex environments, causal discovery can be error-prone, impairing the model's adaptability and leading to suboptimal or even incorrect adaptations. Therefore, it is necessary to see whether CSR can work in a complex environment or task.

- Techniques like causal representation learning for domain adaptation, adaptive policy updates, and structure discovery in RL are not entirely new. Many recent papers explore causal inference within RL and adaptive mechanisms for generalization, such as AdaRL. I cannot tell if this paper's novelty differs from AdaRL.

- Based on my understanding, $\theta$ is trained on the data collected from the target domain. Therefore, how does CSR adapt to an unseen task?

- The self-adaptive strategy, which searches for the optimal structure of causal variables, can be computationally intensive. Can authors clarify the training cost for each step of CSR?

- CSR concentrates on improving the performance of an RL agent with causal representation in dynamic change environments. I am inquisitive about how CSR compares to other curriculum RL methods regarding sample efficiency and final performance, such as ALPGMM.

**Questions:**

Refer to the questions above.

---

> ### Author Response · Authors · 2024-11-20
>
> Thanks a lot for your thoughtful comments. Please see below for our responses to your concerns.
>
> **CSR's potential in noisy and complex environments.**
> Causal representation learning methods, such as CSR, are capable of performing well under complex and noisy conditions for the following reasons:
>
> - **Causal methods, when appropriately designed, hold the potential to outperform non-causal approaches in managing noisy environments [1].**
>   For instance, CIRL [2] employs a causal intervention module, causal factorization module, and adversarial mask module to effectively transform noisy and dependent representations into clean and independent ones, resulting in superior generalization ability. Moreover, learning a causal model uncovers the underlying mechanisms of the data generation process, potentially addressing challenges that non-causal methods may struggle with. For instance, in the case of misalignment [3]—a scenario where an agent’s performance appears aligned with the target goal but is actually influenced by a side goal due to a hidden common cause—causal modeling offers a way to mitigate this issue by identifying causal relationships rather than correlations.
>
> - **Theoretical and empirical evidence supports CSR's robustness.**
>   Theorems 1-3 in our paper provide identifiability guarantees for causal world models under conditions where latent variables or processes have i.i.d. noise terms. Additionally, our experimental results on POMDP tasks, including Atari and CoinRun games, highlight CSR's ability to handle complex environments more effectively than non-causal approaches.
>
> - **Advances in identifiability extend CSR's applicability to challenging scenarios.**
>   Recent developments in causal identifiability have tackled issues such as nonstationary processes [4], time-delayed dynamics [5], heteroscedastic noises [6], and even arbitrary noise distributions [7]. These advancements reinforce the potential of CSR to adapt to diverse and complex environments.
>
> We hope the explanations provided clarify how CSR can leverage causal structures to address such challenges effectively. Thank you again for highlighting this important aspect.
>
> [1] Sanyal et al., 2024. *Accuracy on the wrong line: On the pitfalls of noisy data for out-of-distribution generalisation.* arXiv preprint arXiv:2406.19049.
>
> [2] Lv et al., 2022. *Causality inspired representation learning for domain generalization.* CVPR, 8046-8056.
>
> [3] Di Langosco et al., 2022. *Goal misgeneralization in deep reinforcement learning.* ICML.
>
> [4] Yao et al., 2021. *Learning temporally causal latent processes from general temporal data.* arXiv preprint arXiv:2110.05428.
>
> [5] Yao et al., 2022. *Temporally disentangled representation learning.* NeurIPS, 35, 26492-26503.
>
> [6] Lin et al., 2024. *A skewness-based criterion for addressing heteroscedastic noise in causal discovery.* arXiv preprint arXiv:2410.06407.
>
> [7] Montagna et al., 2023. *Causal discovery with score matching on additive models with arbitrary noise.* CLeaR, 726-751.
>
> **Novelty differs from AdaRL.**
> CSR differs significantly from AdaRL in the following ways:
>
> - **Much More General Application Scenarios**:
>   Unlike AdaRL, which focuses solely on simple distribution shifts for policy adaptation (as stated in line 418), CSR extends to broader scenarios where the spaces may also change across tasks. Details about the distinctions between the two types of scenarios are further given in the Introduction and Appendix C.
>
> - **A Novel Three-step Self-adaptation Strategy**:
>  CSR introduces a three-step strategy specifically designed to address the dynamic nature of the environments it is applied to: (1) Distribution Shifts Detection and Characterization, (2) State/Action Space Expansions, and (3) Causal Graph Pruning. This strategy enables CSR to effectively capture and adapt to environmental changes, a capability not addressed by AdaRL.
>
> - **Theoretical Guarantees**:
>   Prior works, such as AdaRL, lack theoretical support for the identifiability of world models in Eq. (2), whereas CSR not only addresses this gap but also establishes the identifiability of \(\theta\) in linear cases, supported by empirical results in nonlinear scenarios. Furthermore, CSR provides rigorous proofs for the identifiability of newly introduced variables in space expansion settings.
>
> **How does CSR adapt to an unseen task?**
> CSR can be viewed as a method for few-shot policy adaptation. As mentioned in lines 140–141, when a new target task is introduced, CSR collects data from the target domain, specifically $\{\langle o_t, a_t, r_t \rangle \}$, and performs adaptation accordingly. Algorithm 1 in lines 1098-1133 presents the pseudocode for CSR.

---

> > ### Author Response · Authors · 2024-11-20
> >
> > **The training cost of CSR.**
> > It comprises three main steps: (1) distribution shift detection, (2) searching for extended dimensions (if needed), and (3) model (re-)estimation and graph pruning. Notably, step (1) involves comparing $L_{\text{pred}}$ with the threshold $\tau^\star$, and the learning of $D$ and $\theta$ is integrated into the model estimation process in step (3). Therefore, the additional structural and parametric components introduced by CSR do not incur extra training costs compared to Dreamer, except for the searching step (2).
> >
> > As stated in lines 1157-1159, the overall training time aligns with the Dreamer framework. For instance, in Atari experiments, we empirically compare $L_{\text{pred}}$ at step 100k with $\tau^\star$. If space expansion is required, model estimation continues until 400k steps, typically taking one day on an A100 GPU. For the extension strategies, random and deterministic extensions incur negligible costs, while the self-adaptive strategy takes approximately five hours on Atari tasks. However, as mentioned in lines 335–337 and 492–498, the expansion strategy itself is not the focus of CSR. CSR allows flexibility in choosing expansion strategies, and this trade-off should be carefully considered in practical applications.
> >
> > **How CSR compares to other curriculum RL methods?**
> > Thank you for the insightful comment. However, after a thorough investigation and careful review of the relevant curriculum reinforcement learning literature, we find that a comparison with curriculum RL methods may not be directly relevant, as their objectives fundamentally differ from those of CSR.
> >
> > For instance, ALP-GMM, as a curriculum RL method, focuses on "organizing the order in which tasks are presented to a learning agent to maximize its performance on one or several target tasks" [8]. It acts as a teacher, emphasizing how to sample and arrange tasks to create a learning curriculum. In contrast, CSR functions more like a student, tasked with learning and adapting to the given tasks. In fact, CSR aligns more closely with transfer learning methods, which is why we chose methods like AdaRL as our baseline for comparison.
> >
> > [8] Portelas et al., 2020. Teacher algorithms for curriculum learning of deep RL in continuously parameterized environments. CoRL, 835-853.

---

> ### Author Response · Authors · 2024-11-26
>
> Dear Reviewer **PJxf**,
>
> Thank you once again for your time and thoughtful review of our paper. We have addressed your comments, providing clarification on the potential of causal models to handle noisy and complex environments, as well as on the novelty of CSR compared to AdaRL.
>
> Could you please check whether they properly addressed your concern? We would greatly appreciate your feedback. Should there be any further issues, we would be happy to address them. Thank you very much!
>
> Sincerely,
>
> Submission620 Authors

---

> > ### Comment · Reviewer_PJxf · 2024-11-26
> > **Thank you for the response.**
> >
> > I thank the authors for their responses, and I apologize that I didn't reply earlier. It seems that I missed the part about state-action expansion in the original review. This is the reason that I propose the difference between curriculum methods. I have raised the rating. BTW, respond to reviewer jHns, there is a huge gap between the toy environment and AGI. A more complex environment in RL is not equal to AGI, e.g., obstacles in manipulation tasks are challenging.

---

> > > ### Author Response · Authors · 2024-11-26
> > > **Thank you for your valuable feedback!**
> > >
> > > Thank you again for reviewing our rebuttal and for raising the rating! The quality of our paper has greatly improved thanks to your insightful feedback. We will acknowledge all reviewers' valuable contributions in the final version upon publication.

---

### Official Review · Reviewer_vB5A · 2024-11-04

**Soundness:** 2
**Presentation:** 3
**Contribution:** 3
**Rating:** 6
**Confidence:** 4

**Summary:**

This work augments model-based methods such Dreamer with a mechanism for facilitating policy adaption. The method is evaluated on a synthetic task, a modified CartPole with changing dynamics, CoinRun (Procgen) and 5 Atari games with modes and difficulty levels enabled.

**Claims**

C1. The method distinguishes between and addresses two situations: i) distributions shifts ("changes in transition, observation or reward functions") and ii) state/action space expansions ("differences in latent or action spaces")

C2. Causal representation learning (as employed here) uncovers the structural relationships among variables, enabling the agent to autonomously determine in which of the two situations they are in.

**Assumptions**

Identifiability of the model parameters assumes conditional independence between estimated states at different time-steps (Markov condition) and the (stronger?) faithfulness assumption.

Identifiability of the "expanded state space" further assumes linearity, but provides empirical evidence of monotonic correlation between estimated parameters and the ground-truth for simple cases.

**Strengths:**

- the problem tackled here is difficult and highly relevant. Generalization under variations of the underlying dynamics is an area that has shown little progress
- paper is well structured, fairly easy to follow and the description of the method appears to be complete
- I take the results on the toy synthetic task in fig 3a and the modified CartPole experiment to be the most clear evidence in supporting C1 and C2.
- the ablation on the expansion mechanism suggests that indeed the performance gain is not merely correlated with the increased capacity of the estimators but with the quality of the approximation of the underlying causal graph.

**Weaknesses:**

- The paper would benefit from more qualitative analysis in support of C1 and C2. For example graphs showing the evolution of $\mathcal{L}_{\text{pred}}$ and the ratio between the two modes of the algorithm (distribution shift detection and space expansion). What I am going for is whether it would be possible to visualize a correlation between a task and which mechanism is selected during adaption to that task.
- I take the empirical results on Atari to be somewhat weak, with the baseline (Dreamer) often performing statistically insignificantly from the proposed method. Furthermore I would encourage the authors to make a more principled selection of environment when training on a small subset of ALE, for example Atari-5 or Atari-10 games that support tasks [1].
- I have some reservations regarding the way the distinction between the two scenarios is presented. Given that for any number of tasks we could write down the full MDP using a single transition (and reward) matrix, I believe it could be better exemplified not through CoinRun (which is fairly opaque) but maybe through a simple Factored MDP or a structural causal model.

[1] https://arxiv.org/abs/2210.02019

**Questions:**

- Given that we can describe any two tasks as a single MDP with an SCM that contains a task variable that takes values in ${0,1}$ and which can be used to activate/deactivate other state variables, couldn't the method drop the expansion mechanism? Shouldn't learning $D$ and $\theta$ be sufficient to identify the underlying changes in the MDP?
- Is the Dreamer baseline configured to account for the extra learned parameters of CSR (eg. does it have the same state size as CSR eventually reaches)?

---

> ### Author Response · Authors · 2024-11-20
>
> We greatly appreciate your insightful comments and time devoted to our work. We attempt to address all the concerns as follows.
>
> **Visualization of correlation between a task and the adaptation mechanism.**
> Thank you for your valuable suggestion. We have visualized the evolution of $L_\text{pred}$ and the ratio between the two modes of CSR across all four experimental scenarios in Fig. 14. A detailed analysis of these correlations is provided in Appendix D.4.
>
> **Empirical results on Atari.**
> The selection of Atari games in our experiments is guided by a key principle: the tasks must exhibit varying modes and difficulty levels. However, not all games meet this criterion. For example, among the Atari-10 games [1] you referenced, only "Name the Game" and "Bowling" satisfy this requirement. To address this, we carefully selected five games, each with at least two distinct modes and two difficulty levels. Additionally, since our focus is on sequential tasks, we aim to limit training time per task to control costs. This is why we chose games from the Atari 100K set (requiring 1 GPU day per task) instead of Atari 200M (requiring 16 days).
>
> Regarding the comparison with Dreamer, the superior adaptation ability of CSR is particularly evident in its more efficient sample usage and faster convergence to higher scores. As shown in Fig. 16, CSR significantly outperforms other baselines at 100K steps. While this performance gap narrows with more training steps, this trend is entirely expected: with enough training time, any world model (even one trained from scratch) can achieve good scores. This observation actually highlights CSR's higher transfer efficiency, which is one of our key contributions. Furthermore, we have also conducted experiments on Atari-10 games, including "Amidar" (1 mode, 2 difficulties), "Battle Zone" (3 modes, 1 difficulty), "Frostbite" (2 modes, 1 difficulty), and "Qbert" (1 mode, 2 difficulties). The results are presented below.
>
> |              | Amidar   | Battle Zone | Frostbite | Qbert   |
> |--------------|----------|-------------|-----------|---------|
> | Dreamer      | 207.9    | 6805        | 1696.1    | 2566.9  |
> | CSR          | **301.9**| **9080**    | **3562.2**| **3045.1**|
>
> **Distinction between the two scenarios.**
> We have included more detailed examples in Appendix C, along with graphical illustrations, to better clarify the distinction between the two scenarios. Thank you once again for your thoughtful comments!
>
> **Necessity of the expansion mechanism.**
> In our setting, simply learning $D$ and $\theta$ is not always sufficient to ensure the identification of underlying changes. This is because tasks arrive sequentially, and we cannot predict in advance how many variables each task will involve. In other words, it is not feasible to predefine the number of variables, as this approach often turns out to be inadequate.
>
> To illustrate this more clearly, consider an extreme example in the CartPole task. Suppose we set the number of state variables to 4, where these 4 learned variables correspond exactly to the real environment variables: $s_1 = x$ (cart position), $s_2 = \dot{x}$ (cart velocity), $s_3 = \psi$ (pole angle), and $s_4 = \dot{\psi}$ (pole angle velocity). Additionally, the parameters $\theta_1 = m_c$ (cart mass) and $\theta_2 = g$ (gravity) are learned. In this case, for Tasks 1 and 2, the model can easily capture the changes by learning $D$ and $\theta$. However, when a new variable—such as friction $\mu_c$—appears in Task 3, we must expand the state space from $\mathcal{S}_1 \times \mathcal{S}_2 \times \mathcal{S}_3 \times \mathcal{S}_4$ to $\mathcal{S}_1 \times \mathcal{S}_2 \times \mathcal{S}_3 \times \mathcal{S}_4 \times \mathcal{S}_5$, where $s_5 = \mu_c$, in order to adapt to this change. This highlights the necessity of the expansion mechanism in our method.
>
> **Parameter consistency.**
> Yes, we keep the parameters consistent across all baselines for a fair comparison.

---

### Author Response · Authors · 2024-11-20

We are grateful to all reviewers for their efforts and helpful comments regarding our paper. We are encouraged that reviewers **vB5A**, **jHns**, and **qQnp** find our problem setup is interesting and promising; reviewers **vB5A**, **PJxf**, and **qQnp** find our proposed CSR framework is complete and systematic; reviewers **PJxf**, **jHns**, and **qQnp** find our experimental evaluation is broad, and all reviewers find our paper well-structured.

Below is a summary of our responses:

- To reviewer **vB5A**: We have added further experiments and analyses to support C1 and C2. Additionally, we included a simple MDP example in Appendix C to better illustrate the two situations we address.
- To reviewer **PJxf**: We have clarified how CSR differs from AdaRL and demonstrated CSR's potential to handle noisy and complex environments effectively.
- To reviewer **jHns**: We have shown that CSR encodes an interpretable world model and helps mitigate misalignment issues. Furthermore, we have reorganized the experimental evaluations and added explanations in Appendix C to distinguish between the two types of changes you suggested.
- To reviewer **qQnp**: We have included additional implementation details in Appendix D and also in Algorithm 1 to facilitate reproducibility and to inspire further advancements in this area.

Please review our detailed responses to each point raised. We hope that our revisions (the modified text are highlighted in blue) and clarifications satisfactorily address the concerns raised. We thank you again for your valuable time and expertise.

---

### Meta-Review · Area_Chair_h9k6 · 2024-12-19

**Metareview:**

This paper aims to improve RL method’s ability to handle domain shifts and state/action space changes. The core aspect of the proposed method is creating a transition model in a latent space with an expanded state space that maintains some parameters to model the current model dynamics. The approach also handles state and action space changes by shrinking and expanding the latent space and model parameters. A casual modeling approach is used to identify the relationships between relevant parameters. The approach is applied to a modified version of the Dreamer architecture. Experiments demonstrate that the model can adapt to domain shifts and state and action space changes. Theoretical results are also presented that show that the latent space parameters can be identified in the case of linear models.

The modeling of latent space parameters is reminiscent of Bayes adaptive MDPs [1] and other model-based Bayesian RL methods. The authors should review this literature and add references to this line of work where relevant.

[1] Duff, M.O., & Barto, A.G. (2002). Optimal learning: computational procedures for Bayes-adaptive Markov decision processes.

The reviewers all agree this paper is worthy of acceptance. The paper expands on recent adaptive RL methods, most notably regarding the adaptive latent space dimensions. As this successfully addresses the limitations of previous methods, it warrants publication.

**Additional Comments On Reviewer Discussion:**

The reviewers raised many points necessary to clarify the scope of the paper, the relationship to prior works, and the details of the experiments. The authors responded with new results, expanded the discussion, and used simple examples to address the reviewer's concerns. However, the authors should pay close attention to the reviewers' confusion when preparing the manuscript for the camera-ready version.

---

### Decision · Program_Chairs · 2025-01-22

Accept (Poster)